# Sensitivity of Heinrich-type ice-sheet surge characteristics to boundary forcing perturbations

Clemens Schannwell[1], Uwe Mikolajewicz[1], Florian Ziemen[2], and Marie-Luise Kapsch[1]

[1]Max Planck Institute for Meteorology, Bundesstraße 53, 20146 Hamburg, Germany
[2]Deutsches Klimarechenzentrum, Bundesstr. 45a, 20146 Hamburg, Germany

**Correspondence:** Clemens Schannwell (Clemens.Schannwell@mpimet.mpg.de)

**Abstract.** Heinrich-type ice-sheet surges are one of the prominent signals of glacial climate variability. They are characterised as abrupt, quasi-periodic episodes of ice-sheet instabilities during which large numbers of icebergs are released from the Laurentide ice sheet. The mechanisms controlling the timing and occurence of Heinrich-type ice-sheet surges remain poorly constrained to this day. Here, we use a coupled ice sheet-solid earth model to identify and quantify the importance of boundary forcing for the surge cycle length of Heinrich-type ice-sheet surges for two prominent ice streams of the Laurentide ice sheet - the land-terminating Mackenzie ice stream and the marine-terminating Hudson ice stream. Both ice streams show responses of similar magnitude to surface mass balance and geothermal heatflux perturbations, but Mackenzie ice stream is more sensitive to ice surface temperature perturbations, a fact likely caused by the warmer climate in this region. Ocean and sea-level forcing as well as different frequencies of the same forcing have a negligible effect on the surge cycle length. The simulations also highlight that only a certain parameter space exists under which ice-sheet oscillations can be maintained. Transitioning from an oscillatory state to a persistent ice streaming state, can result in an ice volume loss of up to 30% for the respective ice stream drainage basin under otherwise constant climate conditions. We show that Mackenzie ice stream is susceptible to undergoing such a transition in response to all tested positive climate perturbations. This underlines the potential of the Mackenzie region to have contributed to prominent abrupt climate change events of the last deglaciation.

## 1 Introduction

Many hypotheses on what initiates Heinrich-type ice-sheet surges (Heinrich, 1988, henceforth: surges or ice-sheet surges) have been proposed over the last decades (e.g. MacAyeal, 1993; Hulbe et al., 2004; Álvarez-Solas et al., 2011; Bassis et al., 2017). Early theories suggest that the surges are a result of internal ice-sheet oscillations that follow a two-stage pattern in which the ice sheet builds up (binge phase) and subsequently surges (purge phase, MacAyeal (1993)). During the binge phase, ice is frozen to the subglacial material, permitting only ice motion due to internal deformation. As the ice thickens, basal temperatures increase owing to stronger insulation from the cold surface temperatures as well as due to increased strain heating caused by the thicker ice. When the basal temperatures reach the pressure melting point, melwater production lubricates the bed and initiates basal sliding, leading to the purge phase. The surge phase stops when heat generation from internal deformation decreases and cannot maintain basal temperatures at the pressure melting point, resulting in an abrupt stop of basal sliding. Based on

the binge-purge mechanism, a number of studies using ice sheet models of different complexity have been able to reproduce quasi-periodic surge events for synthetic (Payne, 1995; Calov et al., 2010; Feldmann and Levermann, 2017) and real-world geometries (Marshall and Clarke, 1997; Calov et al., 2002; Roberts et al., 2016; Ziemen et al., 2019). The parameterisations that determine the onset of rapid basal sliding however differ between these studies with some using a temperature switch (Marshall and Clarke, 1997; Calov et al., 2002, 2010) and others using a subglacial water availibility switch (Roberts et al., 2016; Feldmann and Levermann, 2017).

The validity of the binge-purge mechanism has since been questioned because most of the documented ice-sheet surges fall into the cold stadials of Dansgaard-Oeschger (DO) cycles (Bond et al., 1993), indicating that a common climatic trigger exists. In the first ice-sheet surge simulation of Hudson ice stream with a 3D ice-sheet model, Calov et al. (2002) reproduced ice-sheet surges during cold DO stadials by applying a weak sliding peturbation with a 1,500 year period to a small region at the snout of the Hudson ice stream, mimicking the effect of a rise in sea level. This revealed the potential of the ocean as possible trigger of ice sheet surges which has also been supported by proxy data showing an ocean warming signal prior to surge events (Marcott et al., 2011). As a result, varying hypotheses identifying the ocean as the key driver for triggering ice-sheet surges have been proposed (e.g. Álvarez-Solas et al., 2011; Marcott et al., 2011; Bassis et al., 2017). Some studies suggest that the increase in ocean temperatures at the ice-ocean interface may have caused the disintegration of a large ice shelf between the Laurentide and Greenland ice sheet (e.g Hulbe et al., 2004; Alvarez-Solas et al., 2013). The sudden removal of the stabilising ice shelf initiates the surge similar to the recent collapse of Larsen B ice shelf and the subsequent speed-up observed of the ice streams in the Antarctic Peninsula region (e.g. Rignot et al., 2004). While this could explain the timing of the surge events during the cold stadials, there is a lack of evidence about the existence of an ice shelf in the reconstructions prior to several recorded surge events (e.g. de Vernal et al., 2000). A more recent theory proposes that warm water intrusions onto the continental shelf in combination with a retrograde sloping subglacial topography trigger widespread ice-sheet retreat (Bassis et al., 2017). Subsequent glacial isostatic uplift cuts off the passage of the warm ocean water onto the continental shelf, leading to a re-advance of the ice sheet. The recent observed widespread retreat of tidewater glaciers in Greenland has also been attributed to the intrusion of warm ocean waters into the fjords (e.g. Straneo and Heimbach, 2013; Murray et al., 2015). Until now, there have only been a couple of attempts to systematically investigate the sensitivity of ice-sheet surges to different boundary forcing perturbations (Calov et al., 2010; Roberts et al., 2016).

A modern-day analogue to ice-sheet surges, albeit on a much smaller scale, are surges of mountain glaciers (e.g. Meier and Post, 1969). Surges of mountain glaciers are also characterised by a build-up phase and a surge phase during which ice velocities of up to 22,000 m/yr have been recorded (Kamb et al., 1985). The concensus is that glacier surges are triggered by internal oscillations (Sevestre and Benn, 2015; Benn et al., 2019). Observations from land-terminating glaciers show that surges are initiated in the upper reaches of the glacier and propagate downstream, while most observations from tidewater glaciers exhibit surge initiation at the terminus and a subsequent up-glacier propagation of the surge (Sevestre et al., 2018). Despite the wide variety of observed surge behaviours, a global analysis of surging glaciers found that they tend to occur only under a subset of global climate conditions and glacier geometries (Sevestre and Benn, 2015), indicating the existence of a unifying theoretical underpinning for these types of surges (Benn et al., 2019).

In this study, the key objective is to identify important external forcings and boundary conditions and quantify their effect on the timing and frequency of ice-sheet surges. We use an ensemble of coupled ice sheet-solid earth simulations and focus our simulations on two regions of the Laurentide ice sheet - the land-terminating Mackenzie ice stream and the marine-terminating Hudson ice stream (Fig. 1). To address these questions, we perform three different types of experiments. The first category investigates the response of time-invariant anomaly peturbations to the surface mass balance (SMB), geothermal heatflux, ice

surface temperature, and sea-level on the surge characteristics. The second category studies the effect of different frequency forcing of the SMB and glacial isostatic adjustment (GIA) on the surge characteristics in comparison to time-invariant forcing. The third type of experiment investigates the interactions between the different surge areas by artificially suppressing surging in one of the surge regions. Our experiments are similar to Calov et al. (2010), but we extend their study by using a coupled ice sheet-solid earth model, perform a richer set of sensitivity experiments, employ a more sophisticated ice-sheet model at higher

horizontal resolution, and simulate real-world ice stream geometries in different glaciological settings.

The paper is structured as follows. Section 2 introduces the coupled model setup, initial state as well as how the idealised climate forcing is generated. In section 3, we first discuss the surge behaviour of the two regions in the Control (Ctrl) simulation with constant forcing, before we detail the differences in the surge mechanism between the land-terminating Mackenzie ice stream and the marine-terminating Hudson ice stream. Subsequently, the sensitivities of the ice sheet surges are presented and

interpreted in context of the different ice stream characteristics. Finally, section 4 summarises the main findings.

## 2 Methods

Proxy data show that most Heinrich and DO events during the last glacial cycle occurred during Marine Isotope Stage 3 (MIS3; North Greenland Ice Core Project members, 2004). Therefore, we use MIS3 climate conditions for our ice sheet-solid earth simulations. By identifying and combining three ice-sheet surge events from a MIS3 simulation with an Earth System Model

(ESM) that includes interactive ice-sheet and solid-earth components, we generate an idealised climate forcing that allows for the explicit modelling of Heinrich-type ice-sheet surges. In the following, we describe the model setup, initial state, and the generation of the idealised composite forcing in more detail.

### 2.1 Model setup and initial state

The coupled ice sheet-solid earth model mPISM-VILMA, consisting of the modified Parallel Ice Sheet Model (mPISM, Ziemen

et al., 2014, 2019) and the global VIscoelastic Lithosphere and MAntle model (VILMA, Martinec et al., 2018) is employed for our simulations. VILMA computes the solid-Earth deformation and the change in relative sea level caused by a redistribution of ice and water. We use VILMA in its 1D configuration, implicitly assuming that the solid-Earth structures only vary with depth, but are otherwise spatially homogeneous. mPISM is employed to compute the ice-sheet response. An earlier version of mPISM has been successfully applied in previous simulations of ice-sheet surges (e.g. Ziemen et al., 2019).

To compute ice velocities, mPISM (Ziemen et al., 2014) uses a simple addition of the Shallow Ice Approximation (SIA) and Shallow Shelf Approximation (SSA) ice velocities. The boundary between grounded ice sheet and floating ice shelves referred

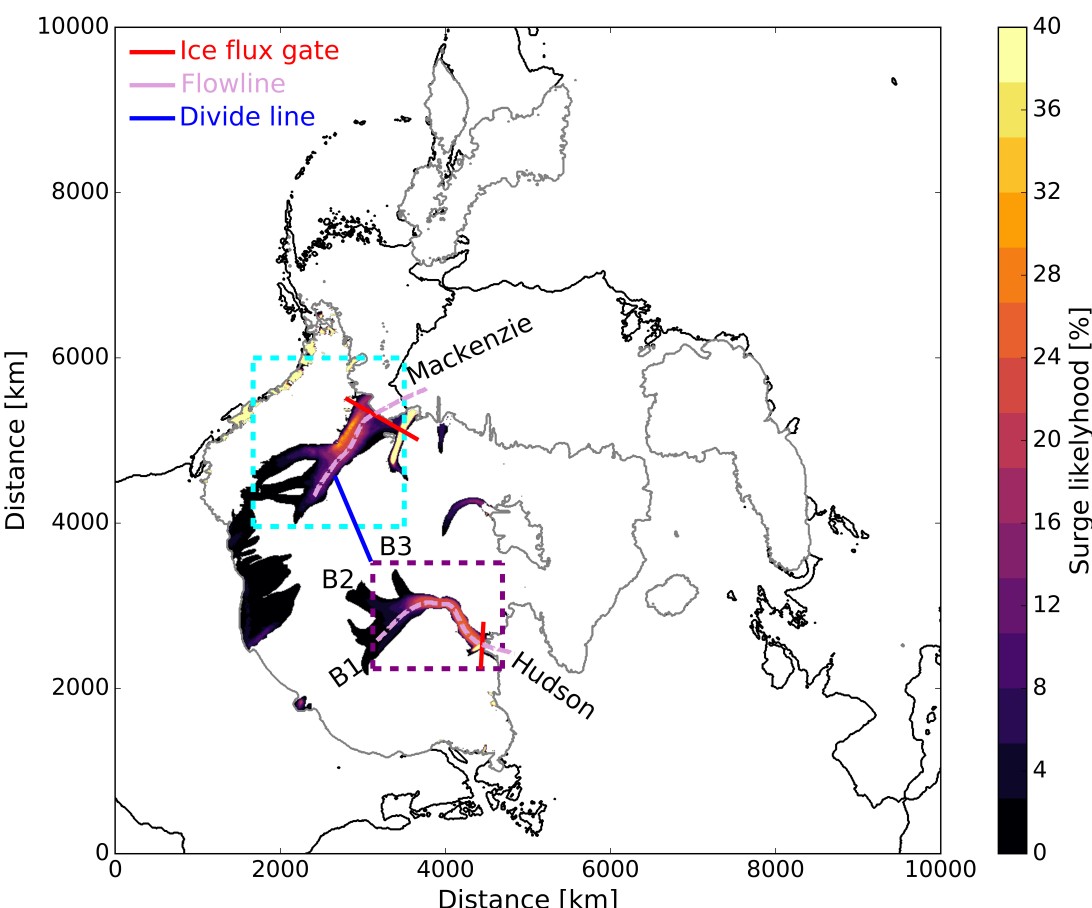

**Figure 1.** Overview of northern hemispheric ice-sheet model domain, highlighting surge regions from the Laurentide ice sheet. Surge likely-hood is calculated from the Ctrl simulation and is the percentage of time that the ice velocity in each cell exceeds 2,000 m yr$^{-1}$. Dashed cyan and purple boxes show areas used for ice volume calculations for Mackenzie and Hudson ice stream, respectively. Pink lines are flowline locations shown in Figs. 4 and 6. Red lines show flux gates across which ice flux for the respective ice stream is calculated. Solid blue line shows line approximately perpendicular to the drainage basin divide between Hudson and Mackenzie drainage basins. Labels B1, B2, and B3 depict different surge branches of the Hudson ice stream.

to as grounding line is determined from the flotation condition without additional flux conditions imposed. Basal melt under floating ice shelves is calculated from ocean temperature and ocean salinity following the three-equation method of Holland and Jenkins (1999). The ocean temperature and salinity fields are provided by the Max Planck Institute for Meteorology ESM

(MPI-ESM). Because MPI-ESM does not resolve the cavities under ice shelves, we extend salinity and temperature fields to

ice shelves through extrapolation. This 2D extrapolation is based on temperature and salinity averages between ∼200-400 m depth of the ocean model. The differences in the model setup that go beyond changes in the release versions of MPI-ESM and PISM, and the methods employed in the coupling, are described in the following.

The current version of mPISM is based on PISM version 0.7.3. As in Ziemen et al. (2014), we modified the enthalpy advection to use the c-grid method that is also utilised for the mass advection. We also added the model fields and methods necessary for the coupling to the other model components. For the interpolation of the SSA velocities onto the c-grid, a weighted mean with 70% upwind and 30% downwind velocity is used. To improve the sliding behaviour necessary for the ice-sheet surges, we spread 50% of the basal heating effect from the sliding in each cell over the four neighbour cells. This leads to a faster surge propagation and shorter surge periods that is more in accordance with reconstructions.

The surge behaviour in the model is strongly influenced by how basal sliding is parameterised. We apply a nonlinear Weertman-type friction law at the base of the ice sheet, where ice is in contact with the subglacial topography that links the basal shear stress $\boldsymbol{\tau}_b$ to the basal sliding velocity $\boldsymbol{u}_b$ of the form:

$$\boldsymbol{\tau}_b = -\tau_c \frac{\boldsymbol{u}_b}{u_0^q |\boldsymbol{u}_b|^{1-q}}. \tag{1}$$

where $q$ is the basal sliding coefficient – set to 0.25 in all simulations, $\tau_c$ is the yield stress, and $u_0$ is the threshold velocity at which the basal shear stress has the same magnitude as $\tau_c$. We set $u_0$ to 70 m/yr for all simulations. The yield stress $\tau_c$ is computed through the Mohr-Coulomb criterion

$$\tau_c = tan(\phi) N_{till}. \tag{2}$$

Here, $\phi$ represents the till friction angle that is parameterised based on bedrock elevation. In our simulations, the friction angle varies linearly from 15 to 30 for bedrock elevations between -300 m and 400 m and remains constant at the upper or lower limit otherwise. The effective pressure ($N_{till}$) is determined from the ice overburden pressure and the till saturation as described in Bueler and van Pelt (2015). Following Calov et al. (2002) and Ziemen et al. (2019), basal sliding is further enhanced in regions where sediment is present based on the sediment map of Laske and Masters (1997). This is implemented in the model by scaling $\tau_c$ from equation 2 by a factor 1/100 for areas where sediment thickness exceeds 0.2 m and keeping it unchanged elsewhere. The choice of the scaling factor and sediment thickness threshold were selected manually, but are motivated by the rationale that rapid sliding should be restricted to areas with suffciently thick sediment cover at the bottom of the ice sheet. For the forcing of the temperature equation, we apply a time and space invariant geothermal heat flux of 42 mW m$^{-2}$, if not stated otherwise. The geothermal heatflux is specified at the bottom of a 1 km thick bedrock model that is located below the ice sheet. The initial state of our mPISM-VILMA simulations was taken at 36 ka before present from a MIS3 simulation with MPI-ESM asynchronously coupled to mPISM and VILMA. This time was selected because the global ice volume during this time period remained stable apart from the cyclic ice-sheet surge events. The MPI-ESM model system was adapted for long-term simulations in comparison to the version presented in Mauritsen et al. (2019) and accounts for changes in the land-sea mask, ocean bathymetry (Meccia and Mikolajewicz, 2018), and river directions (Riddick et al., 2018) induced by changes in the glacial configuration. A detailed description of the MPI-ESM setup without the interactive ice-sheet and solid-earth components can

be found in Kapsch et al. (2021, 2022). For the MIS3 simulation, MPI-ESM was additionally coupled to the solid-earth model VILMA and to a bi-hemispheric setup of mPISM, employing a 10 km resolution in the Northern Hemisphere and a 15 km resolution in the Southern Hemisphere. The coupling between MPI-ESM and mPISM-VILMA was performed asynchronously with an acceleration factor of 10 for the MPI-ESM component.

In the present study, we use a northern hemispheric setup of mPISM (Fig. 1) with a horizontal mesh resolution of 10 km. The coupling of mPISM and VILMA is performed every 100 model years and includes the exchange of the ice thickness distribution and GIA between the two models. Since VILMA is a global model, it requires a global ice thickness distribution as input. As we focus on the Laurentide ice sheet, we keep the ice thickness distribution of all other ice sheets constant through time. The simulations are integrated for a total of 85,000 years of which the first 20,000 years are considered model spin-up and only the last 65,000 years are used for analysis.

## 2.2 Generation of idealised climate forcing

The generation of an idealised forcing is motivated by two main objectives that aim to facilitate the analysis of the performed simulations. The first objective is to improve the signal-to-noise ratio of the underlying forcing and reduce the influence of forcing variability that is not related to ice-sheet surges. The second objective is to create a forcing that exhibits a minimal temporal trend and can be repeated as many times as desired. The latter requires a smooth transition from one forcing cycle to the next (Fig. 2b).

To improve the signal-to-noise ratio, we generate a composite forcing of three surge events from the aforementioned MIS3 simulation with the asynchronously coupled MPI-ESM/mPISM/VILMA model system. We base the selection of characteristic surges on the Hudson ice stream, which is the most prominent surge region of the Laurentide ice sheet. The peak discharge of the three surges is manually matched and subsequently averaged (black line, Fig. 2a). To ensure that the transition of subsequent forcing cycles are smooth, we apply a sinusoidal weight function to the mean forcing of the three events. The weight function is applied such that each point of the final forcing is a linear combination of two points from the mean forcing. The red dots in Fig. 2a illustrate the two points that are used to determine the final forcing at the center point (year 3,250 in Fig. 2b). For the remainder of the points located in the purple box in Fig. 2a, the weight points are then progressively moved closer together during which their weights become more similar until they are neighbouring points with effectively equal weights (points located exactly on purple box in Fig. 2a). This process is carried out separately for the increasing and decreasing branch of the forcing cycle. An example of two cycles of the resulting final forcing for a one-dimensional variable field is presented in Fig. 2b. For two-dimensional forcing fields, the same process is applied for each individual grid point. This method provides a forcing with a periodicity of 6,500 years that consists of the climate feedback signal of an ice-sheet surge event and can be repeated as many times as desired. We use this procedure for the atmospheric and ocean forcing. The atmospheric forcing is calculated and downscaled from fields of the MPI-ESM/mPISM/VILMA simulation using an Energy Balance Model (EBM, Kapsch et al., 2021) and consists of the ice surface temperature and SMB fields. The ice surface temperature hereby corresponds to the temperature of the lowest layer in a snow model (Kapsch et al., 2021), restricting ice

surface temperature to values below freezing. Ocean forcing contains ocean temperature and salinity fields extracted from the ocean component of the MPI-ESM/mPISM/VILMA system.

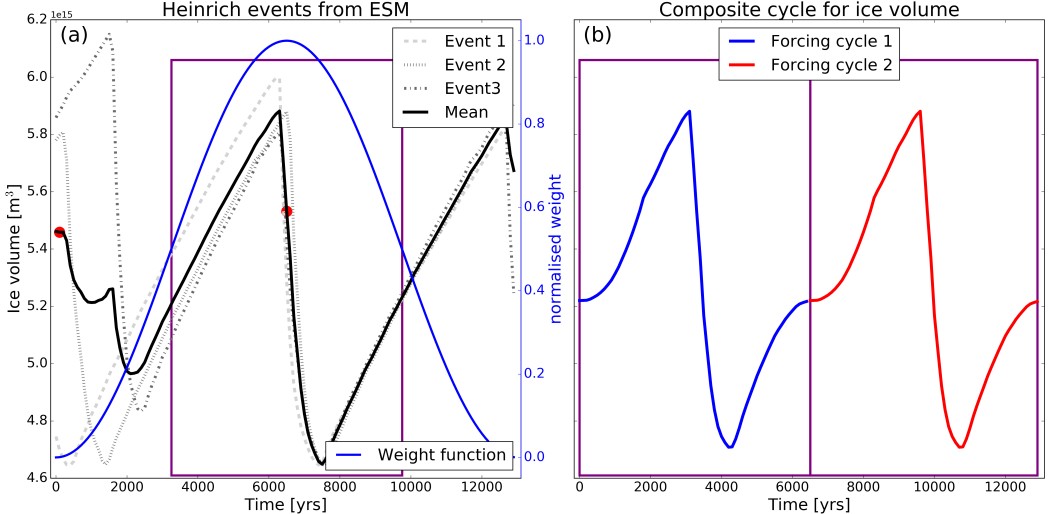

**Figure 2.** (a) Construction of composite forcing from individual ice-sheet surge events of Hudson ice stream from a MIS3 simulation with the MPI-ESM/mPISM/VILMA model system. Red dots indicate the start points for the application of the weighting function from which the final forcing is generated. (b) shows two cycles of ice volume changes from the composite forcing after the application of the weight function. There is a smooth transition from one forcing cycle to the next. Purple box in (a) corresponds to two forcing cycles shown in (b) after the application of the weight function.

## 3 Results and discussion

In the next sections, we will present and discuss the ensemble of ice sheet-solid earth simulations (Table 1). We use the control experiment Ctrl as baseline experiment. Ctrl uses a time constant forcing, which is derived by taking the mean forcing of the cyclic forcing (Cf) outlined in Section 2.2. We use Ctrl to first provide an overview of the general surge behaviour of the Mackenzie and Hudson ice stream in our simulations. Subsequently, we use Ctrl to show that Hudson and Mackenzie ice stream exhibit a different surge initiation behaviour that is most likely caused by their distinct glaciological setting. We then explore the impact of the magnitude and variability of the boundary forcing on the surge cycle of the two ice streams. Finally, we investigate whether the surge behaviour of each individual ice stream is affected by surges of the other ice stream.

### 3.1 Surge behaviour of Hudson and Mackenzie ice streams in the control simulation

Both ice streams exhibit periodic high ice discharge events that we identify as ice-sheet surge events. The surges are a robust feature for most simulations presented here and have previously been shown for the Hudson ice stream in Ziemen et al. (2019).

**Table 1.** List of all perturbation experiments including type and magnitude of forcing. Abbreviations stand for control (Ctrl), geothermal heatflux (Geo), surface mass balance (Smb), ice surface temperature (St), sea level (Sl), ocean temperature (Ot), cyclic forcing (Cf), frozen (Froz), Hudson (Hud), and Mackenzie (Mac). The - and + signs represent negative and positive perturbations. Experiment type refers to time constant anomaly forcing (1), time varying frequency forcing (2), and extreme time constant anomaly forcing to suppress surges (3).

| Experiment Name | Forcing type | Perturbation | Type of perturbation (magnitude) | Experiment type |
|---|---|---|---|---|
| Ctrl | Mean | Constant forcing | None | 1 |
| Geo++ | Mean | Higher geothermal heatflux | Anomaly (+42 mW m$^{-2}$) | 1 |
| Geo+ | Mean | Higher geothermal heatflux | Anomaly (+21 mW m$^{-2}$) | 1 |
| Geo-- | Mean | Lower geothermal heatflux | Anomaly (-42 mW m$^{-2}$) | 1 |
| Smb++ | Mean | Higher SMB | Anomaly (+100 kg m$^{-2}$yr$^{-1}$) | 1 |
| Smb+ | Mean | Higher SMB | Anomaly (+50 kg m$^{-2}$yr$^{-1}$) | 1 |
| Smb-- | Mean | Lower SMB | Anomaly (-100 kg m$^{-2}$yr$^{-1}$) | 1 |
| St++ | Mean | Higher surface temperature | Anomaly (+5°C) | 1 |
| St+ | Mean | Higher surface temperature | Anomaly (+2.5°C) | 1 |
| St-- | Mean | Lower surface temperature | Anomaly (-5°C) | 1 |
| Sl+ | Mean | Higher sea level | Anomaly (+10 m) | 1 |
| Sl- | Mean | Lower sea level | Anomaly (-10 m) | 1 |
| Ot+ | Mean | Higher ocean temperature | Anomaly (+2°C) | 1 |
| Cf | Cyclic | Cyclic forcing | Frequency (6500 years) | 2 |
| CfSmb+ | Cyclic | Higher SMB frequency | Frequency (factor 2 faster) | 2 |
| CfSmb- | Cyclic | Lower SMB frequency | Frequency (factor 2 slower) | 2 |
| CfGeo+ | Cyclic | Higher GIA frequency | Frequency (factor 2 faster) | 2 |
| CfGeo- | Cyclic | Lower GIA frequency | Frequency (factor 2 slower) | 2 |
| CtrlFrozHud | Mean | Extreme negative geothermal heatflux | Anomaly (-1000 mW m$^{-2}$) | 3 |
| CtrlFrozMac | Mean | Extreme negative geothermal heatflux | Anomaly (-1000 mW m$^{-2}$) | 3 |

The modelled high ice velocities and large velocity gradients during the surges pose a formidable challenge for the model numerics. However, in all presented simulations, solutions to the stress balance and ice thickness evolution equation always converge, confirming the robustness of our results. The surge frequency between the two regions differs significantly. The mean interval between ice-sheet surges is 4,700 years for the Mackenzie ice stream and 7,200 years for the Hudson ice stream

(Fig. 3). This is in good agreement with proxy data for the Hudson ice stream which suggests a periodicity between 7,000-13,000 years (Heinrich, 1988). Overall, the Hudson ice stream exhibits larger peak discharge and the surges tend to be longer than for the Mackenzie ice stream. It is worth noting that the two surge regions appear to oscillate independently from each other (see supplementary video). There is little variability for Hudson beyond the big surge events. In comparison, Mackenzie ice stream shows smaller intermittent increases in ice discharge between the major surge events (Fig. 3).

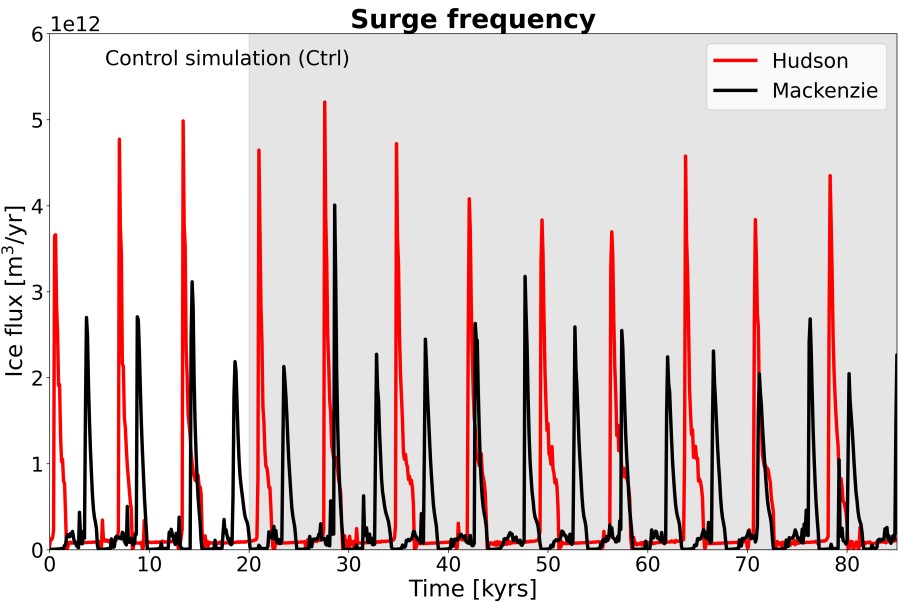

**Figure 3.** Timeseries of the ice flux through the flux gates presented in Fig. 1. Peak discharges define ice-sheet surge events. Grey shading highlights analysis period.

## 3.2 Differences in surge mechanism between Mackenzie and Hudson ice stream

All ice-sheet surges in our model setup are the result of internal ice sheet oscillation that follow the broad theory of the binge-purge mechanism (MacAyeal, 1993). The surge sequence is described by three main stages: the quiescent phase, the pre-surge phase, and the surge phase. However, the initiation and the propagation mechanisms differ for the Mackenzie and Hudson ice stream. For the marine-terminating Hudson ice stream a small region close to the terminus is at pressure melting point (warm-based) during the quiescent phase, while the remaining parts further upstream are well below the pressure melting point (cold-based, Fig. 4a,b). As the ice sheet thickens and enters the pre-surge phase, the surface slope and hence the driving stress start to increase. This slope increase is largest in the area of the transition from warm-based to cold-based subglacial conditions (Fig. 4c,d) resulting in enhanced warming and an expansion upstream of the warm-based region making more subglacial water available to lubricate the bed. When this region becomes extensive enough, the large ice velocity gradient at the bottom warm-cold boundary triggers an activation wave during which mechanical heat dissipation leads to an abrupt and widespread warming and lubricating of the ice stream (Fig. 4e,f). In about 100 years, this activation wave reaches the upper region of the drainage basin (Fig. 5). Almost all of the Hudson ice stream surges are characterised by contributions from three different surge branches (B1, B2, B3 in Fig. 1). The branches are almost exclusively activated in sequence, starting with branch B1 before branches B2 and B3 surge (Fig. 5b). The next branch is activated when the previous branch ceases to exist. This is the case as soon as the active branch cannot draw down more ice to fuel the surge. The total surge phase lasts between 1,500-2,000 years before the advection of cold ice from upstream leads to a shutdown of the surge.

These results indicate that the surge initiation for the marine-terminating Hudson ice stream is driven from the ice-stream

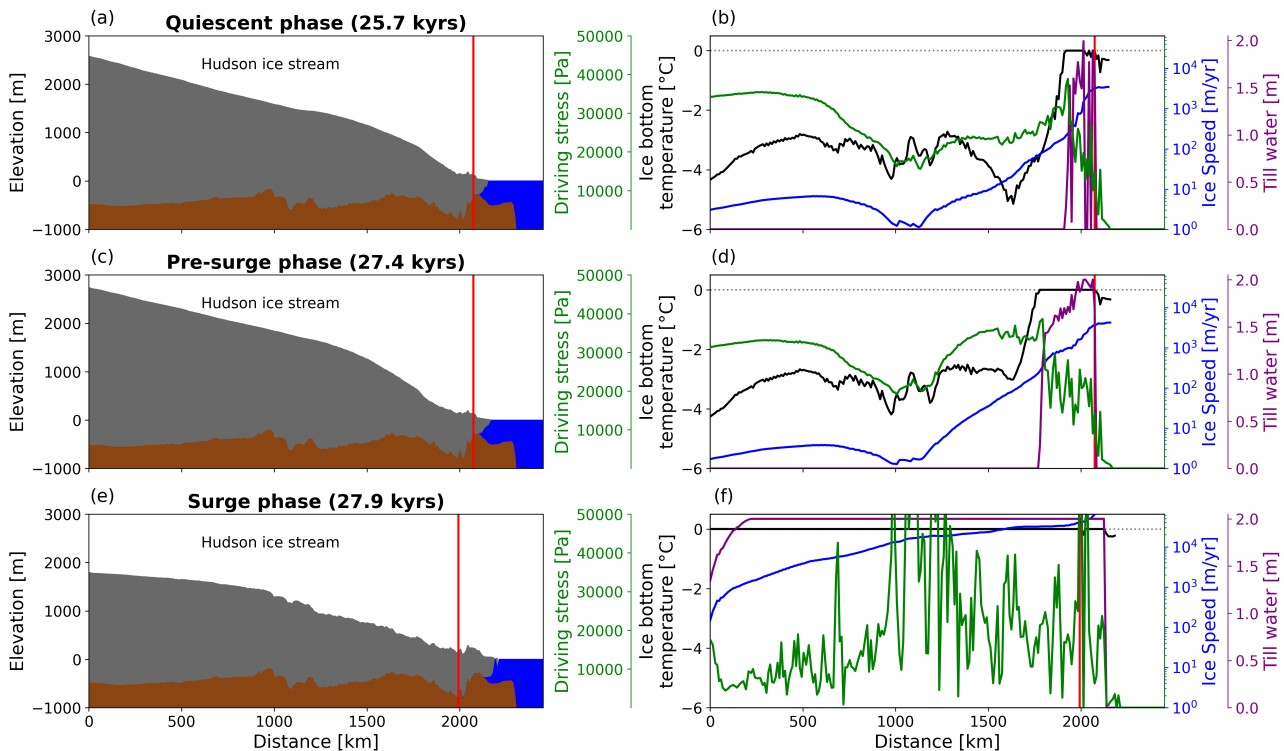

**Figure 4.** The three distinct phases during a canonical surge cycle for Hudson ice stream along the flowline in Fig. 1. Left panel shows ice sheet geometry. Right panel shows pressure adjusted bottom temperature and driving stress as well as ice speed and the thickness of basal water layer. Red vertical line approximates grounding-line position.

front and propagates upstream. A similar surge mechanism has been described in previous modelling attempts (Payne, 1995; Calov et al., 2002; Feldmann and Levermann, 2017) and has also been observed for present-day surging tidewater glaciers (e.g Sevestre et al., 2018). In contrast, surges of the land-terminating Mackenzie ice stream are initiated further upstream and propagate downstream. During the quiescent phase, the terminus of the Mackenzie ice stream is well below the pressure melting point and therefore provides a barrier that has to be overcome for a surge event to take place (Fig. 6a,b). As the ice sheet thickens during the pre-surge phase, driving stress increases and results in a warming of the ice-bedrock interface. This leads to smaller-scale accelerations and ice sheet advances when narrow regions in the lower part of the ice stream reach the pressure melting point (Fig. 6c,d). However, as the ice stream is not preconditioned to surge yet, these accelerations are short-lived. Only when a large enough portion of the upstream part of the ice stream reaches the pressure melting point and the base of the ice sheet is lubricated a full surge develops (Fig. 6e,f). The resulting surge front then propogates downstream. This behaviour is more akin to the initially proposed binge-purge mechanism (MacAyeal, 1993) and closely resembles the behaviour of present-day surging mountain glaciers (e.g Sevestre et al., 2018). Further evidence for the similarity between the surging of

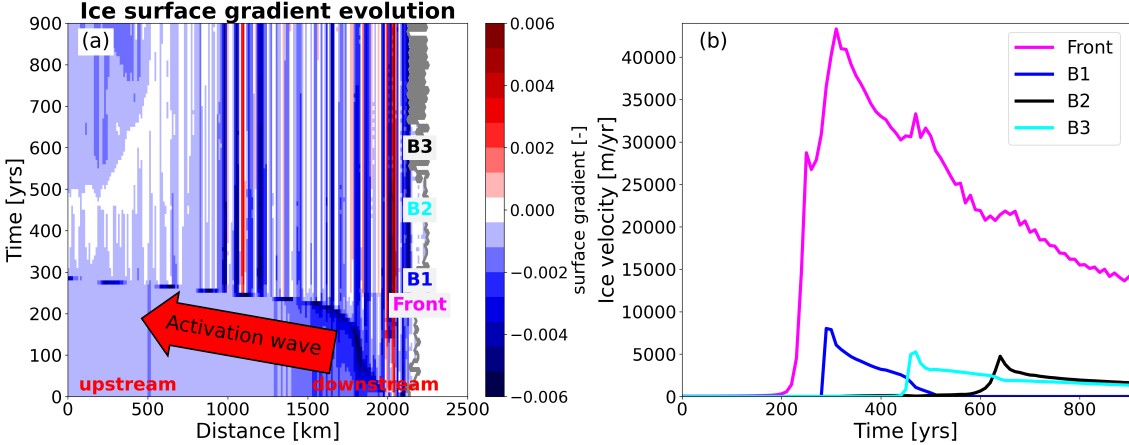

**Figure 5.** (a) Hovmoeller diagram of Hudson ice stream along the flowline (Fig. 1) during a surge event. The surface gradient captures the rapid upstream expansion of surge activation front. Grey line shows marine margin. (b) shows activation of different surge branches of Hudson ice stream for the same surge event. Points for velocity sampling in (b) were manually selected. Labels in (a) refer to timing of surge activation of different branches (B1, B2, and B3) shown in (b).

land-terminating mountain glaciers and the surge behaviour of Mackenzie ice stream is provided by the observation that these surge events are commonly associated with ice-sheet advance (Fig. 6e,f), a characteristic that is entirely absent from Hudson ice stream surges. The warmer basal temperatures further upstream on Mackenzie ice stream could be a result of the generally warmer air temperatures for this region caused by the lee effect of the Cordilleran ice sheet and the insulation effect that is provided by the thicker ice sheet. Compared to the Hudson ice stream (1,500-2,000 years), the surge duration is shorter (1,000-

1,500 years). The shutdown of the surge follows the same pattern as for the Hudson ice stream. The absence of a well defined subglacial trough in the Mackenzie drainage basin results in more variable locations of the Heinrich-type ice-sheet surges, a behaviour that is also supported by reconstructions of Mackenzie ice stream locations from proxy data (Margold et al., 2014). This underlines that the glaciological as well as the climatic setting alter the surge initiation process even though the governing surge mechanism for both ice streams is identical.

### 3.3    Sensitivity of surge cycle to anomaly forcing

In the following, we investigate the effect of boundary forcing perturbations on the surge cycle characteristics of the two ice streams. The time constant anomalies are added to the time constant baseline forcing from Ctrl. We test responses to perturbations of the SMB, ice surface temperature, geothermal heatflux, sea level, and ocean temperature. For all but the ocean

temperature forcing, a positive and negative anomaly experiment was performed (Table 1). To sample the parameter range more densely, we also performed an additional positive intermediate anomaly experiment for the variables SMB, ice surface temperature, and geothermal heatflux. All perturbations experiments are compared to Ctrl.

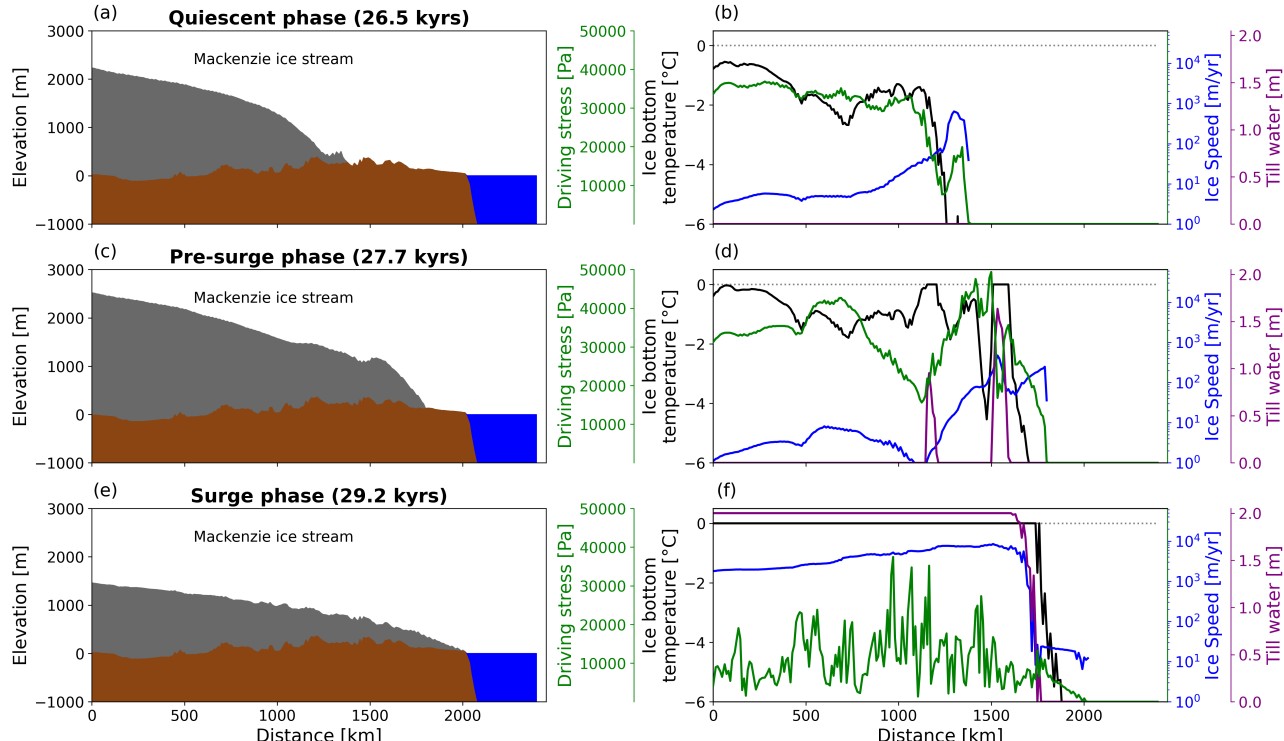

**Figure 6.** Similar to Fig. 4 but for Mackenzie ice stream.

In the positive (Smb++) and negative (Smb--) SMB experiments, the SMB is varied by $\pm 100$ kg m$^{-2}$ yr$^{-1}$. This corresponds to a perturbation that is $\sim 3$ times and $\sim 6$ times larger than the maximum SMB variation in Cf for the Mackenzie and Hudson drainage basins, respectively (Fig. A2). However, while this perturbation may seem large, the tested range is much smaller than the SMB spread that can be expected from temperature and precipitation values shown in the Paleoclimate Modelling Intercomparison Project - Phase 4 (PMIP4) LGM ensemble (Fig. A1) and also falls into the range of what previous studies have used (Calov et al., 2010; Bassis et al., 2017; Feldmann and Levermann, 2017).

The SMB perturbations significantly alter the surge cycle length. For the Hudson ice stream, the surge cycle length is reduced by $\sim 1,700$ years for Smb++ to 5,500 years and lengthened by $\sim 3,100$ years to 10,800 years for Smb--. A similar trend, albeit different in magnitude, is observed in Smb-- for the Mackenzie ice stream, where the surge cycle length increases by $\sim 2,800$ years to 7,500 years (Fig. 7a, b). For Smb++, the ice-sheet oscillations in the Mackenzie region cease to exist and turn this region into a persistent ice stream. For the Hudson ice stream, a shorter surge cycle length in Smb++ is associated with lower peak discharge values for each event, resulting in less ($\sim 32\%$) ice volume discharge. Moreover, the surge behaviour switches from single peaked events to double peaked events $\sim 58$ kyrs into the simulation. This switch results in a wider spread of simulated surge periods as double peaked surges exhibit a longer surge period than single peak events. In comparison, Smb-- leads to a longer surge cycle with initially larger peak discharge values that decrease in the latter stages of the simulation. The

opposite is observed for Mackenzie ice stream, where a longer surge cycle in Smb-- results in smaller peak discharge values and consequently a reduction in discharged ice volume of $\sim$21% per surge. The intermediate simulation Smb+ also switches from a single peaked surge behaviour to a double peaked surge regime for Hudson ice stream, albeit $\sim$12 kyrs later than in Smb++ (Fig. B1a). However, in terms of the ice volume discharge and surge period, Smb+ and Smb++ provide similar results (Fig. 8b), indicating that the response to SMB perturbations of the Hudson ice stream is non-linear. For the Mackenzie ice stream, Smb+ also results in the transition to a persistent ice stream, but the transition occurs later in comparison to Smb++ (Fig. B1b).

In the geothermal heatflux perturbation simulations (Geo++, Geo--), geothermal heatflux is varied by 42 mW m$^{-2}$, meaning geothermal heatflux ranges from 0-84 mW m$^{-2}$ in our perturbation simulations. This choice is in a similar range to that of geothermal heatflux maps of present-day North America (Lucazeau, 2019). Perturbations to the geothermal heatflux of this magnitude lead to similar changes in the average surge cycle length than SMB perturbations for the Hudson region. For Hudson ice stream, surge cycle length is shortened by $\sim$1,900 years to 5,300 years in Geo++ and lengthened by $\sim$1,800 to 9,000 years for Geo--. In Geo++, there is a change in the surge characteristic for Hudson ice stream with a shift towards double peaked surges (B1 and B3 in Fig. 1). A continued addition of heat to the system turns the Hudson ice stream eventually into a persistent ice stream with smaller scale oscillations. The longer surge cycle length in Geo-- results in larger peak discharge and discharged ice volume ($\sim$21%) per surge event. For the Mackenzie area, the positive anomaly almost immediately turns this region into a persistent ice stream, while Geo-- results in an extension of the surge cycle by $\sim$1,400 years to 6,100 years. Here, the longer surge cycle length is also associated with increased peak discharge and slightly elevated discharged ice volume values (Fig. 8b). The switch to double peaked surge events also occurs in the intermediate Geo+ simulation for Hudson ice stream. The characteristic that the double peaked surge events are spaced further apart than the double peaked events simulated in the SMB experiments complicate the computation of the surge cycle length, making this a less reliable metric and explains the increased variability in that variable (Fig. 8a). In contrast to Smb+, Geo+ shows a nearly linear response in the discharged ice volume. The fact that Geo+ as well as Geo++ switch from a single peak surge behaviour to a double peak surge behaviour and Geo++ eventually enters the state of persistent streaming suggests that this switch can be interpreted as a precursor for the transition to a persistent streaming state. For the Mackenzie ice stream, Geo+ also results in the transition to a persistent ice stream after three surges (Fig. B1d).

In the ice surface temperature simulations (St++, St--), ice surface temperature is varied by 5°C. This is well below what proxy data suggest as temperature variations during a DO cycle (e.g. Kindler et al., 2014) and also much smaller than the spread shown in the PMIP4 LGM ensemble (Fig. A1). The mean response to ice surface temperature perturbations is smaller in magnitude than geothermal heatflux and SMB perturbations for the Hudson ice stream and similar in magnitude to the geothermal heatflux perturbation for the Mackenzie ice stream. For the Hudson ice stream, St++ results in a decrease of the surge cycle length of $\sim$500 years to 6,700 years. The surge cycle length is extended by the same amount in St--, leading to a surge cycle length of 7,700 years. The Mackenzie ice stream again turns into a persistent ice stream for St++, while St-- results in an increase in surge cycle length of $\sim$1,200 years to 5,900 years, more than twice as large as the Hudson ice stream response (Fig. 7e, f). Large variations in peak discharge for the Mackenzie region are evident in St--, with large peak discharge events

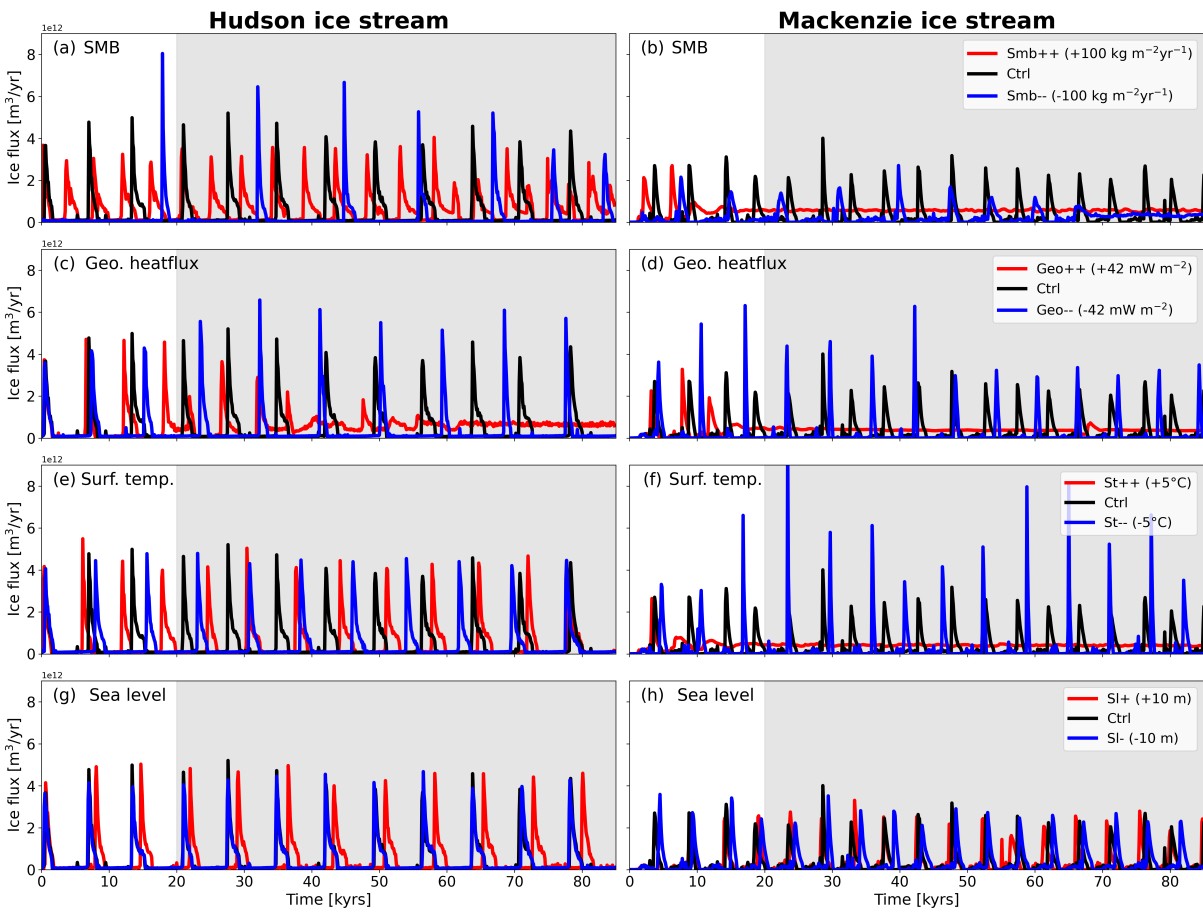

**Figure 7.** Similar to Fig. 3 but for Hudson (left) and Mackenzie (right) ice stream separately. The panels show response to (a,b) SMB, (c,d) geothermal heatflux, (e,f) ice surface temperature, and (g,h) sea level perturbations. Grey shading highlights analysis period.

occuring more frequent. The higher peak discharge values are associated with a mean increase in discharged ice volume of ~44% for each event. A similar correlation between changes in ice surface temperature and surge cycle length was found for individual models in ISMIP HEINO, but this correlation was not robust across all participating models (Calov et al., 2010). The response of the intermediate St+ perturbation of the Hudson ice stream is perhaps the most linear of the three tested variables. For discharged ice volume and surge period, St+ is located approximately halfway in between the Ctrl and St++ simulation (Fig. 8a). For the Mackenzie ice stream, St+ also results in the transition to a persistent ice stream after one surge event (Fig. B1d).

In the sea level simulations (Sl+, Sl-), sea level is changed by 10 m. This is within the range of sea-level rise estimates for Heinrich-type ice sheet surges inferred from proxy data (e.g. Hemming, 2004) and is ~8% of the sea-level change from the Last Glacial Maximum to present-day (Lambeck et al., 2014). Overall, sea-level perturbations have very little bearing on the surge cycle length (Fig. 7g, h). For Hudson ice stream, Sl- shows identical surge timing and magnitude as Ctrl. For Sl+, the

surge interval is slightly lengthened over the first three surge cycles before the system reaches a new equilibrium and oscillates at the same frequency as Ctrl, but surges are shifted by about 800 years (Fig. 7g). Peak discharge and discharged ice volume per surge remain practically unchanged in comparison to Ctrl. For Mackenzie ice stream, Sl- and Sl+ show very similar surge behaviour to Ctrl (Fig. 7h). This is expected as Mackenzie ice stream is land-terminating and therefore should be unaffected by changes in sea level. Mackenzie shows no change in surge cycle length, peak discharge, and discharged ice volume per surge event. However, the timing of the surges in Sl- and Sl+ differs slightly in comparison to Ctrl (Fig. 7h). We attribute these small changes to the high sensitivity of the Mackenzie ice stream to any system perturbations and its more variable surge behaviour in the absence of a well-defined subglacial trough.

In an additional simulation (Oc+), we also investigate the effect of warmer ocean temperatures for the marine-terminating Hudson ice stream. For this, the ocean temperature is increased by 2°C in the Oc+ simulation. However, Hudson ice stream only forms small floating ice tongues in our simulations. As a consequence, there is no change in surge cycle length for the Hudson ice stream to increased ocean temperatures (Fig. B2).

Based on the three different common boundary forcings that were tested, we find that SMB and geothermal heatflux both result in changes of similar magnitude with respect to surge cycle length and discharged ice volume for the individual ice streams. However, overall the effect of the SMB on the surge cycle length is larger for Hudson ice stream than for Mackenzie ice stream. Moreover, the St simulations show that Mackenzie ice stream is more sensitive to changes in ice surface temperature than the Hudson ice stream (Fig. 8). We attribute these differences to differences in their respective glaciological and climatic settings. The Hudson ice stream area is characterised by ∼8-10 °C colder ice surface temperatures. The colder temperatures place the Hudson ice stream in a SMB regime that is dominated by accumulation with no ablation area (Fig. A2). This system reacts more sensitively to changes in SMB than an ice stream where the lower reaches experience significant amounts of ablation, as is the case for the Mackenzie ice stream. Due to very negative ($<$-500 kg m$^{-2}$ yr$^{-1}$) ablation values for the Mackenzie region, the applied perturbations ($\pm$100 kg m$^{-2}$ yr$^{-1}$) are relatively small in comparison to the Hudson ice stream, resulting in smaller changes to the surge cycle length.

The perturbation experiments highlight that the transitional behaviour and the parameter space, under which ice-sheet oscillations are possible, are different for Mackenzie and Hudson ice stream. Hudson ice stream shows a precursor for the transition to a persistent streaming with a switch from single peaked to double peaked surges which is not simulated for Mackenzie ice stream. While Hudson ice stream only shows non-oscillatory behaviour in one of the perturbation simulations, Mackenzie ice stream enters the regime of persistent ice streaming for almost all positive perturbation simulations. We attribute this enhanced sensitivity of Mackenzie ice stream to a different surge mechanism in comparison to Hudson ice stream and the warmer climate conditions in this region, which is close to the climatic threshold that still supports ice-sheet surges. This is corroborated by the observation that the distribution of present-day surging glaciers also appears geographically bounded by key climatic parameters such as surface temperature and precipitation (Sevestre and Benn, 2015). The consequences of a regime shift from oscillations to a persistent ice stream are significant in terms of ice volume and associated sea-level rise over longer time scales (∼10,000 years). In the case of Hudson ice stream, such a regime shift results in a volume loss of 30% over the entire simulation length in comparison to the constant baseline simulation (Ctrl), adding up to a total of 3.8 m sea-level equivalent (Fig. 9a).

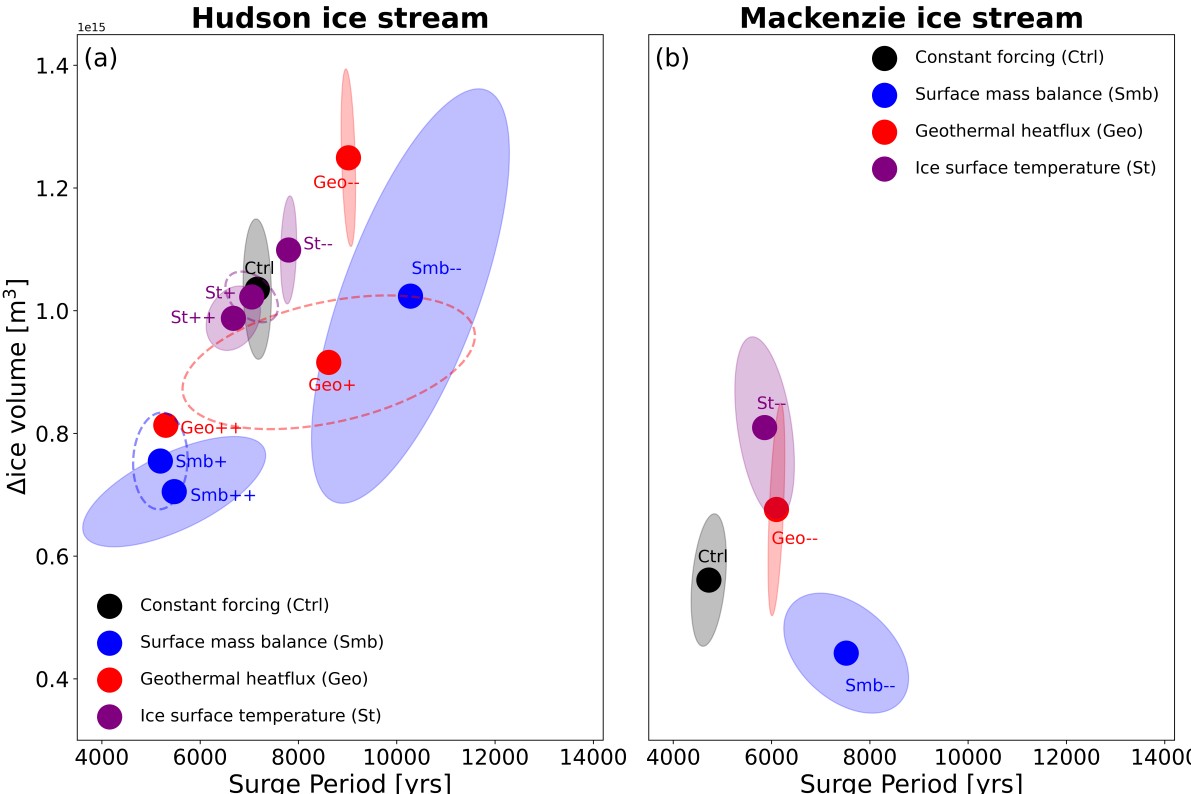

**Figure 8.** Ice volume change over surge period for (a) Hudson ice stream and (b) Mackenzie ice stream. Ice volume change is computed by taking the difference between maximum and minimum ice volume across each surge cycle. The surge period is defined as time elapsed between subsequent ice discharge peaks. Ellipses show ±1 standard deviation, but are only plotted if more than four events were simulated. Sea level perturbations are omitted because they plot on top of Ctrl experiment for both regions. Positive perturbation simulations are missing in (b) because of an insufficient number of surges in these simulations due to the transition to a persistent ice stream.

For Mackenzie ice stream, the ice volume response is faster but smaller, primarily because its drainage basin is smaller in size
and stores less ice. However, the increase in global sea level due to a regime shift of the Mackenzie ice stream still amounts to up to 1.8 m (Fig. 9b).

The applied perturbations in the experiments are well below the spread shown by the PMIP4 ensemble. This indicates that the sensitivities presented in this study may have important ramifications for abrupt climate events during the MIS3 period as well as the progression of the last deglaciation. As Mackenzie ice stream is more sensitive to any climate warming signal than Hudson ice stream, it is more suceptible to have undergone the dynamical switch from an oscillatory system to a steady streaming system during the last deglaciation. Because of this enhanced sensitivity, we argue that it is also likely that this transition occurred in the earlier phase of the last deglaciation, potentially contributing to prominent abrupt climate change events of the last deglaciation such as Meltwater Pulse 1A (14.5 ka, Weaver et al. (2003); Gregoire et al. (2012)) or the onset

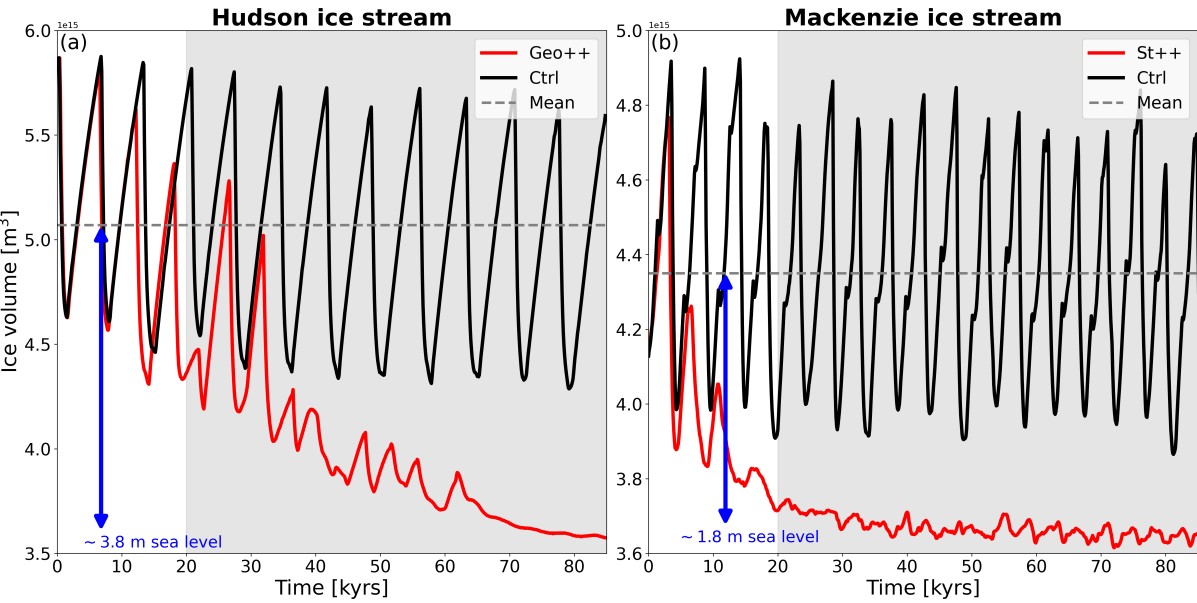

**Figure 9.** Comparison of ice volume evolution between oscillatory and persistent ice stream regime for the (a) Hudson and (b) Mackenzie ice stream. Grey shading highlights analysis period.

of the Younger Dryas cold period. A previous study already indicated the potential contribution of Mackenzie ice stream to
the onset of the Younger Dryas (Tarasov and Peltier, 2005). Possible consequences of such a dynamical switch go beyond the
freshwater release into the ocean and sea-level response. A continuous loss of ice results in a lowering of the surface topogra-
phy, potentially affecting the timing of the separation of the Laurentide and Cordilleran ice sheets, as well as changes in river
directions, which can result in a redistribution of meltwater from the Laurentide ice sheet (Kapsch et al., 2022).

### 3.4   Sensitivity of the surge cycle to forcing frequency

We have investigated the surge cycle response to time invariant anomaly perturbations. Here, we study the effect of different
forcing frequencies on the surge cycle. The cyclic forcing contains the climate feedback signal of the atmosphere and ocean
component of the MPI-ESM/mPISM/VILMA model system during a surge event from the MIS3 period. The main objective is
to investigate if this type of forcing is able to influence the timing of ice-sheet surges. For this, we performed a simulation with
the cyclic forcing derived in Section 2.2 (Cf). To study the effect of single forcings, we also performed additional simulations in
which the frequency of the SMB was increased by a factor of two (Cfsmb+) or decreased by a factor of two (Cfsmb-). A similar
kind of frequency perturbation was performed for the response of the glacial isostatic adjustment (GIA, Cfgeo+, Cfgeo-). In
these simulations, mPISM was run without VILMA and the GIA forcing fields from Cf were either contracted or stretched by
a factor of two. The choice of the two tested single forcings is motivated by the result that the response of GIA perturbation
was not investigated and the SMB highly affected the surge cycle length in the anomaly experiments.

First, we compare Cf with Ctrl. Cf has a forcing frequency of 6,500 years, while Ctrl has time constant forcing. In comparison to the anomaly perturbation simulations, the different forcing frequencies have very little effect on the surge cycle length. For Hudson ice stream, the surges are almost identical between the simulations. This is of little surprise for Ctrl and Cf because the forcing of Cf is based on the natural surge frequency of Hudson ice stream and, even if Hudson ice stream responds to different forcing frequencies, should not result in a different surge cycle length. More variability is evident for the Mackenzie ice stream, but no clear trend emerges. That Mackenzie ice stream responds more sensitively to system perturbations is in accordance with the findings from the anomaly perturbation experiments. The cyclic climate forcing does not affect the timing of the surges in our simulations. This means that the timing of the surges does not conincide with the timing when the cyclic climate forcing suggests that a surge should occur (Fig. 10, right panel).

The results are similar for the simulations in which only the frequency of the SMB or GIA forcing is varied. Differences are very small for the Hudson ice stream, regardless of the parameter that is being varied. Again, there is more variability in the response of Mackenzie ice stream, but there is no clear trend towards shorter surge cycles for a higher frequency forcing (Fig. 10, right panel). A potential reason for this increased variability is the absence of a well-defined subglacial trough in the Mackenzie region of the ice sheet.

Several studies have shown that ice-sheet surges can be phase locked (e.g. Calov et al., 2002; Kaspi et al., 2004; Mann et al., 2021). Phase locking refers to the observation that two weakly coupled autonomous oscillators with different natural frequencies can synchronise if they both are exposed to an external forcing that is strong enough to overcome the natural frequency of the two oscillators (Kaspi et al., 2004). In the present case, the two oscillators are the ice sheets and the climate system coupled through the boundary forcing. Phase locking has been used to explain the occurrence of ice-sheet surges during cold stadials of DO cycles, suggesting that a common climate trigger is determining the timing of ice-sheet surges (Kaspi et al., 2004; Mann et al., 2021). Typical reported timescales for the manifestation of phase locking are less than 10 kyrs (Kaspi et al., 2004; Mann et al., 2021). In our simulations, we do not observe phase locking between the boundary forcing and surge activity, regardless of the parameter and forcing frequency (Fig. 10). Instead, both surge regions are unaffected by the boundary forcing and continue oscillating at their natural frequency. This indicates either that the internal oscillation mechanism is not affected by the frequency of the boundary forcing or that the climate signal present in the cyclic forcing from the MIS3 simulation is insufficient to cause phase locking. A potential explanation for the lack of phase locking is the inability of the ESM, from which the climate forcing for the presented simulations was taken, to produce DO-like climate oscillations. This means that shorter frequency oscillations like the 1,500 year long DO cycle are not included. It also suggests that the forcing magnitude is most likely underestimated because it only consists of the climate feedback signal induced by ice-sheet surges, but lacks the component of the DO events.

## 3.5 Telecommunication between surge regions

Phase locking has also been used to explain synchronised ice-sheet surges either from different locations of the Laurentide ice sheet or from different northern hemispheric ice sheets (Calov et al., 2002; Kaspi et al., 2004). In this section, we investigate if the coupling between the two surge regions exists and how large of an effect surges of Mackenzie ice stream have on surge

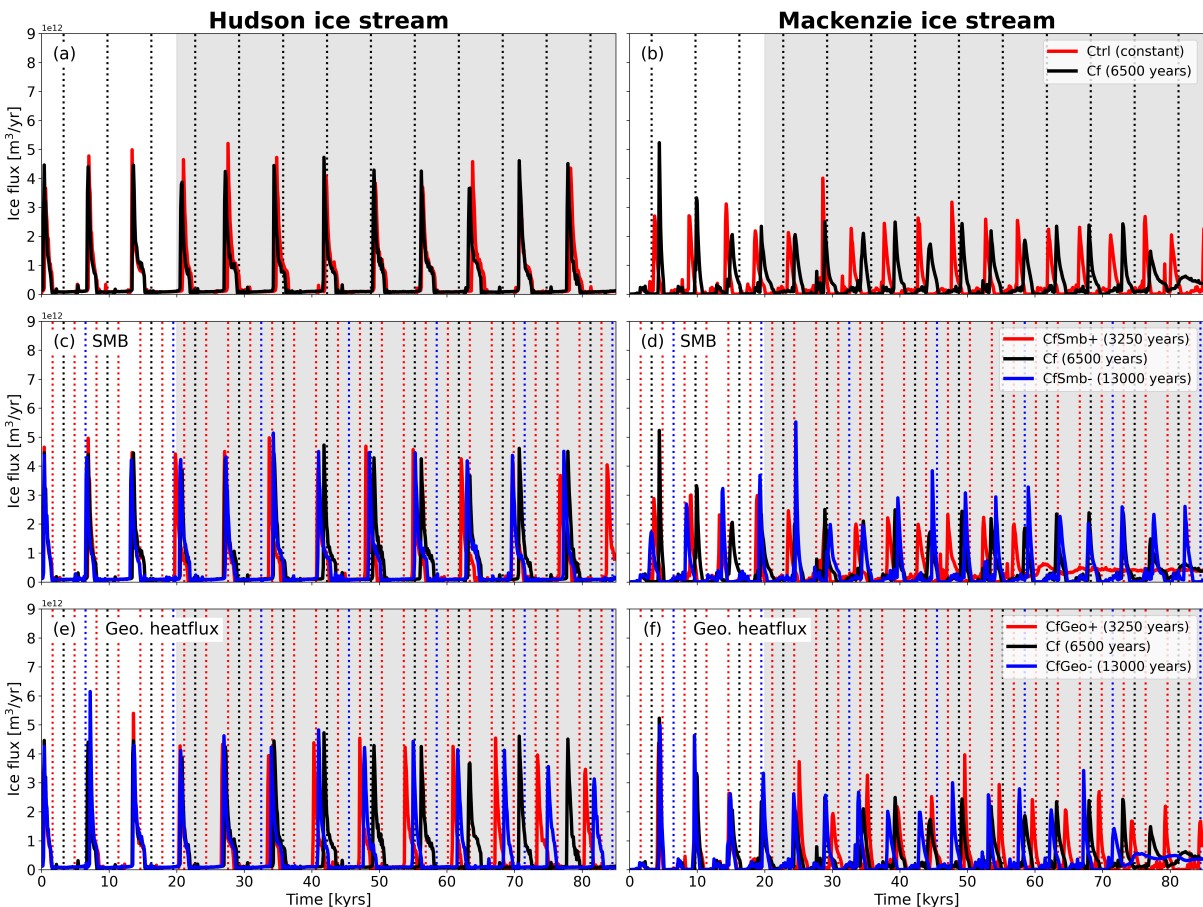

**Figure 10.** Similar to Fig. 3 but for Hudson (left) and Mackenzie (right) ice stream. The panel show response to (a,b) constant and cyclic forcing, (c,d) different SMB forcing frequencies, and (e,f) different glacial isostatic adjustment frequencies. The dashed vertical lines indicate when surge should occur according to the derived forcing. Numbers in parentheses in the legends indicate forcing frequency. Grey shading highlights analysis period.

characteristics of the Hudson ice stream and vice versa. To do this, we perform additional simulations in which we use constant boundary forcing similar to Ctrl, but artificially suppress surging in one of the two surge regions. Surges are suppressed by applying a negative geothermal heatflux that keeps the ice-sheet base well below pressure melting point (CtrlFrozHud, Ctrl-FrozMac).

The surge regions are dynamically connected because they share a common ice divide. The position of the divide marks the highest point of the Laurentide ice sheet and also marks the boundary of the drainage basins of Mackenzie and Hudson ice stream. In Ctrl, the divide migrates by ∼200 km over a time span of ∼1,500 years between surge events. During an ice-sheet surge, the dynamic thinning that accompanies the increase in ice discharge results in a migration of the ice divide and a growing of the drainage basin area. On a much smaller scale, this has also been modelled for Antarctic ice rises (e.g. Schannwell

et al., 2019). However, the long-time mean position of the divide remains unchanged (Fig. 11). This means that the average drainage basin area of the surge regions does not change and, hence, changes in surge characteristics are not to be expected. In CtrlFrozHud and CtrlFrozMac the ice stream that remains active experiences a lasting increase in drainage basin size. In both simulations, the divide migrates by ∼400 km into the drainage basin of the inactive region. The larger drainage basin offers the potential for a change in surge characteristics because it allows for more accessible ice volume that potentially could be mobilised during a surge event. This leads to a change in surge behaviour for the Hudson ice stream, but there is little change for the Mackenzie ice stream, despite a similar increase in drainage basin area. The reason for the different response is that the added drainage basin area is never activated in subsequent surges of the Mackenzie ice stream. This is due to the fact that the added drainage basin area for Mackenzie ice stream remains well below pressure melting point during the subsequent surge events. In contrast, the Hudson ice stream surges now propagate much further upstream, which switches the single peaked surge regime to a double peaked surge regime (Fig. 11a), also observed in the Geo++ simulation. The double peaked surges are characterised by surges from two different branches of the ice stream (see B1 and B3 in Fig. 1). In between the ice discharge maxima, elevated ice velocities and ice discharge are maintained. Only when the second branch starts to surge does the other ice stream cease to exist. These results show that the drainage basin size is an indicator for the potential magnitude of ice discharge, but only under the assumption that the added ice volume is actually activated during surges.

The position of the main ice divide of the Laurentide ice sheet has also implications for the atmospheric circulation, which is not considered in our model system. Because it marks the highest point of the ice sheet, it provides a barrier for atmospheric flow. It has been shown in previous studies using ice-sheet reconstructions that different ice sheet heights strongly affect the climate response (e.g. Ullman et al., 2014; Bakker et al., 2020; Kapsch et al., 2022). Hence, it is reasonable to conclude that ice divide migration of several hundred kilometer east or west is likely to affect the climate response.

As suggested by some proxy data (e.g. Grousset et al., 2000) and simulated in simple box models (Kaspi et al., 2004), synchronisation of surge events from different ice sheets or different regions of the same ice sheet can occur through phase locking. The scenario of synchronised surges from the Laurentide ice sheet has important implications for the global climate response as more meltwater is added simultaneously at multiple geographic locations to the ocean. The importance of the amount of added freshwater and the location of the freshwater input for the global ocean response has been highlighted by a variety of freshwater hosing experiments (e.g. Maier-Reimer and Mikolajewicz, 1989; Schiller et al., 1997; Stouffer et al., 2006; Smith and Gregory, 2009; Lohmann et al., 2020). With our standard setup employed here, synchronised surges cannot be reproduced through phase locking. The only avenue to simulate surge events that appear synchronised from the Hudson and Mackenzie ice stream with our model setup is to combine different boundary forcings that result in similar surge cycle lengths. For example, the Geo-- simulation for Mackenzie ice stream and the St++ simulations for Hudson ice stream show such a behaviour (Fig. 12a). However, this necessitates that the surge events are already sychronised from the start of the simulation (Fig. 12b).

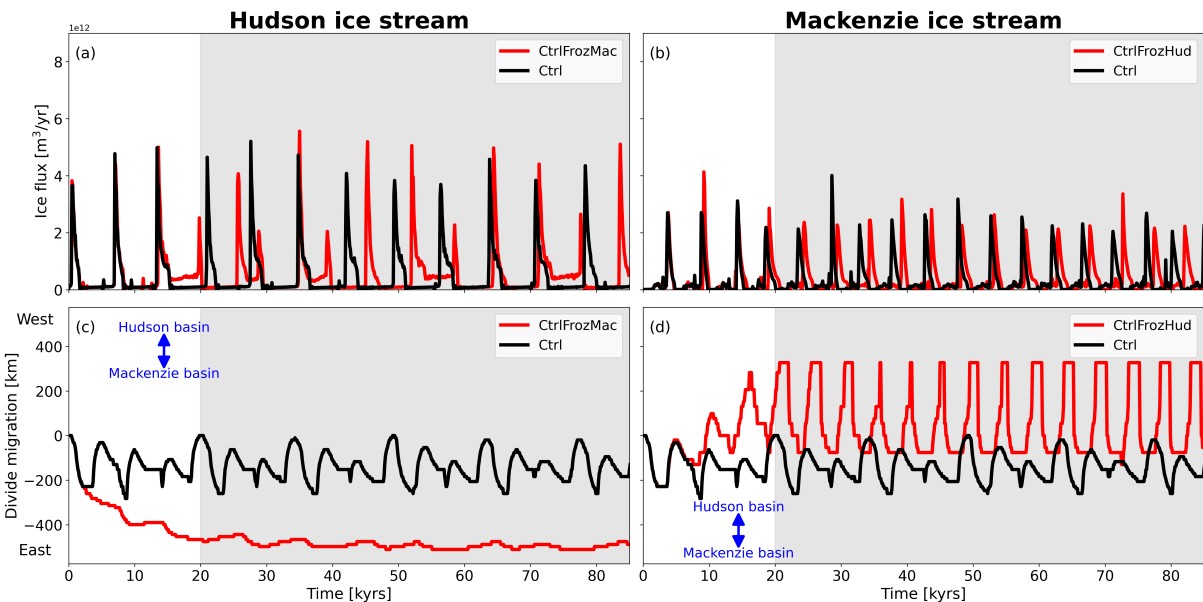

**Figure 11.** Differences induced by artificially suppressing surge events in one of the surge regions. Upper panel (a,b) shows timeseries of ice flux through the flux gates (Fig. 1). Lower panel (c,d) shows temporal evolution of ice divide position (Fig. 1).

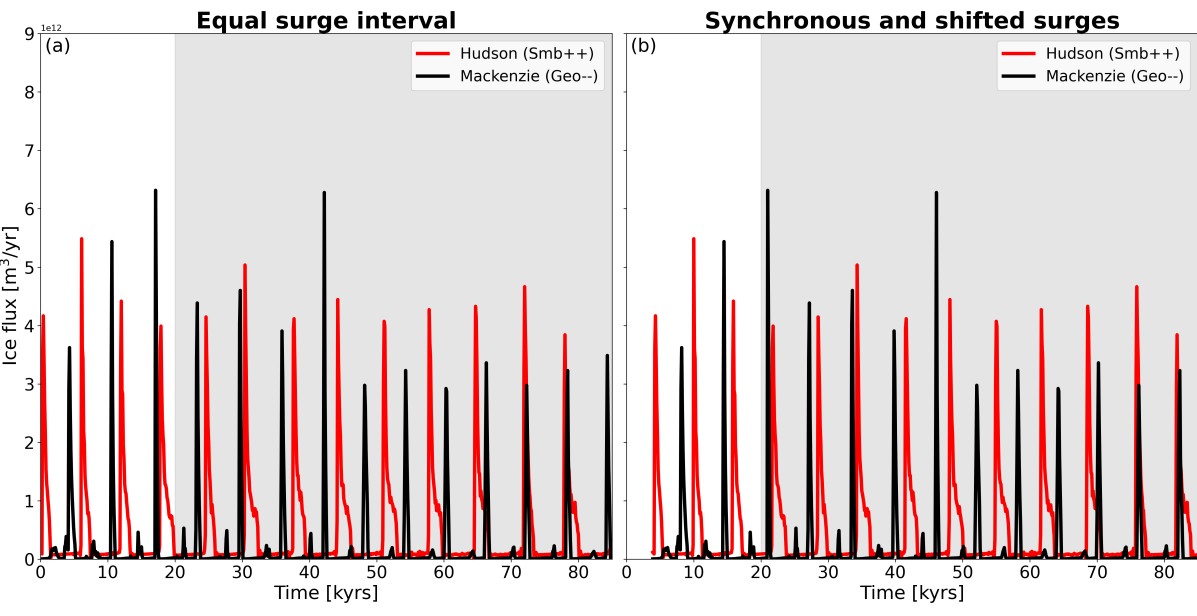

**Figure 12.** Timeseries of the ice flux through the flux gate (Fig. 1) for Hudson and Mackenzie ice stream showing (a) equal surge interval and (b) synchronous surges.

## 4 Conclusions

Our simulations identify different surge characteristics for the land-terminating Mackenzie ice stream and the marine-terminating Hudson ice stream. The surges of Hudson ice stream are initiated at the ice stream front and propagate upstream. This surge onset closely resembles the surge behaviour observed for present-day tidewater glaciers in Svalbard (Sevestre et al., 2018). In contrast, the surge behaviour of Mackenzie ice stream is more akin to the initially proposed binge-purge mechanism (MacAyeal, 1993) with surge initiation occurring further upstream and a subsequent downstream propagation of the surge. Despite their
different glaciological and climatic settings, both ice streams show responses of similar magnitude to perturbations to the SMB and the geothermal heatflux. However, Mackenzie is more sensitive to surface temperature changes, a fact that we mainly attribute to the warmer climate conditions in this area. Perturbations to the ocean temperature and sea-level forcing as well as different forcing frequencies have little effect on the surge cycle length. This is in contrast to findings from earlier studies (Calov et al., 2002; Alvarez-Solas et al., 2013; Bassis et al., 2017). The likely reasons for the insensitivity of Hudson ice stream
to warmer ocean temperatures is that in our simulations Hudson ice stream never forms an extensive ice shelf that could provide a stabilising force.

Both regions are affected by surges of the other region through the migration of the ice divide that their drainage basins share. However, the increase in drainage basin size only leads to a change in surge behaviour towards double-peaked events for the Hudson ice stream. Even though the peak ice discharge does not change, the length of the surge is extended, leading to a
445 higher ice volume loss for each surge event. The surge behaviour remains unchanged for the Mackenzie ice stream because the additional available ice volume is not activated during the surges. Synchronised surges from the Mackenzie and Hudson ice stream cannot be modelled under the control boundary forcing, but under certain boundary forcing combinations such as Smb++ for Hudson and Geo-- for Mackenzie ice stream, surges can appear synchronised because they exhibit similar surge cycle lengths.

The simulations also highlight that Mackenzie ice stream is close to a parameter regime under which ice-sheet oscillations cannot be maintained. The dynamical switch from oscillations to persistent ice streaming can result in a volume loss of up to 15% and 30% in comparison to constant climate conditions for Mackenzie and Hudson ice stream, respectively. If both regions enter the ice streaming parameter space at a similar time, an additional 5.6 m of sea-level equivalent over ∼50 kyrs could be added to the global ocean. The extreme sensitivity of Mackenzie ice stream to any positive perturbation in the anomaly
experiments underlines the potential of Mackenzie ice stream to have contributed to prominent abrupt climate change events during glacial-interglacial transitions, such as the Younger Dryas cold period.

## Appendix A: SMB and surface temperature variations from MPI-ESM and PMIP4 ensemble

In this section, we justify the chosen ranges of the surface temperature and SMB anomaly experiments by comparing them to the spread shown in the Paleoclimate Modelling Intercomparison Project - Phase 4 (PMIP4) LGM simulations as well as the SMB spread exhibited by MPI-ESM for one cycle of the idealised forcing. For the comparison with the PMIP4 ensemble, we approximate the model spread by computing the difference between ensemble maximum and ensemble minimum. As proxy for surface mass balance, we select precipitation. This assumes that there is negligible amount of melt at the LGM and that all precipitation falls as snow. Precipitation and surface temperature fields show a much larger spread in the PMIP4 ensemble than the range tested in our anomaly simulations. Mean temperature and mean precipitation spreads for the Hudson ice stream and for the Mackenzie ice stream are 22 K, 265 kg m$^{-2}$ yr$^{-1}$ and 14 K, 334 kg m$^{-2}$ yr$^{-1}$, respectively (Fig. A1). In comparison to the SMB variations simulated based on the idealised forcing generated from MPI-ESM, the maximum values tested are ∼3 times larger for Mackenzie ice stream and ∼6 times larger for the Hudson ice stream (Fig. A2). Both tests illustrate that our chosen ranges are within a realistic envelope.

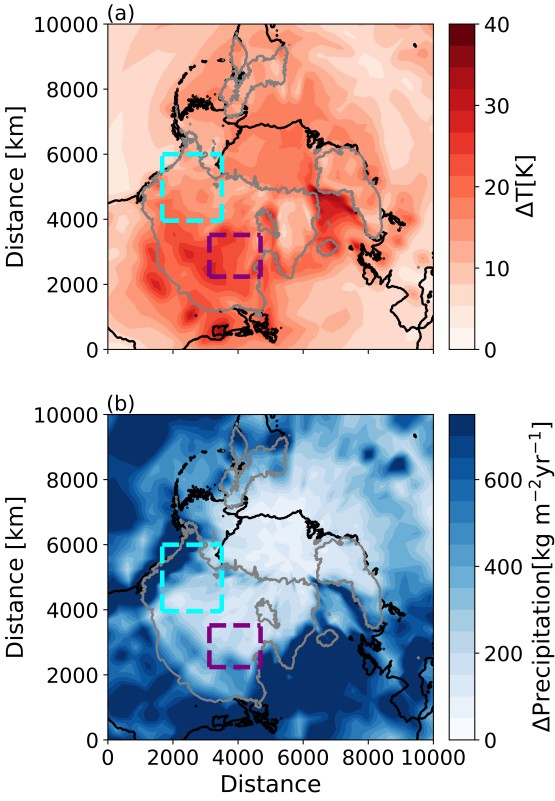

**Figure A1.** Difference between PMIP4 ensemble maximum and ensemble minimum for (a) surface temperature (b) precipitation. Cyan and purple dashed boxes show Mackenzie and Hudson area respectively. Grey lines outline ice sheet extent.

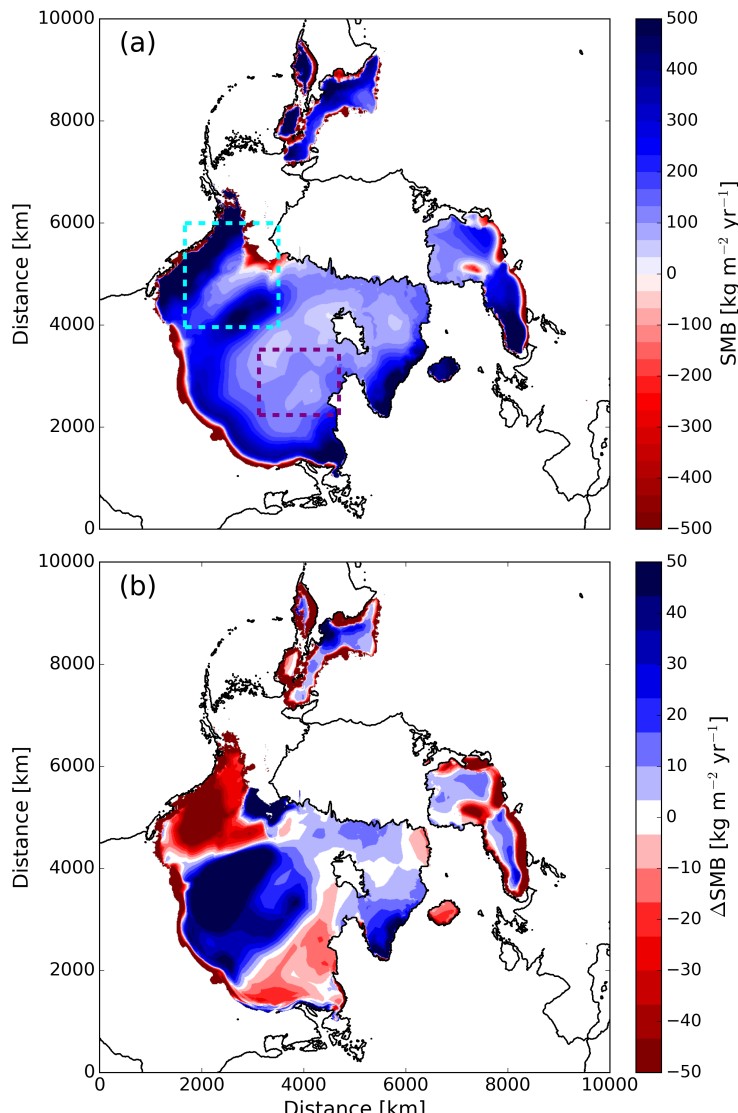

**Figure A2.** (a) Mean SMB field for one cycle of the idealised forcing. (b) Difference in SMB before and after a surge event for one cycle of the idealised forcing. Cyan and purple dashed boxes in (a) show Mackenzie and Hudson area respectively.

## Appendix B: Additional anomaly perturbation experiments

In this section, we present results of additional simulations which are intermediate positive perturbation simulations of the variables SMB, geothermal heatflux, and ice surface temperature (Fig. B1) as well as a simulation were the effect of an increase in ocean temperature on the surge cycle was tested (Fig. B2). The ocean perturbation experiment is restricted to the marine terminating Hudson ice stream.

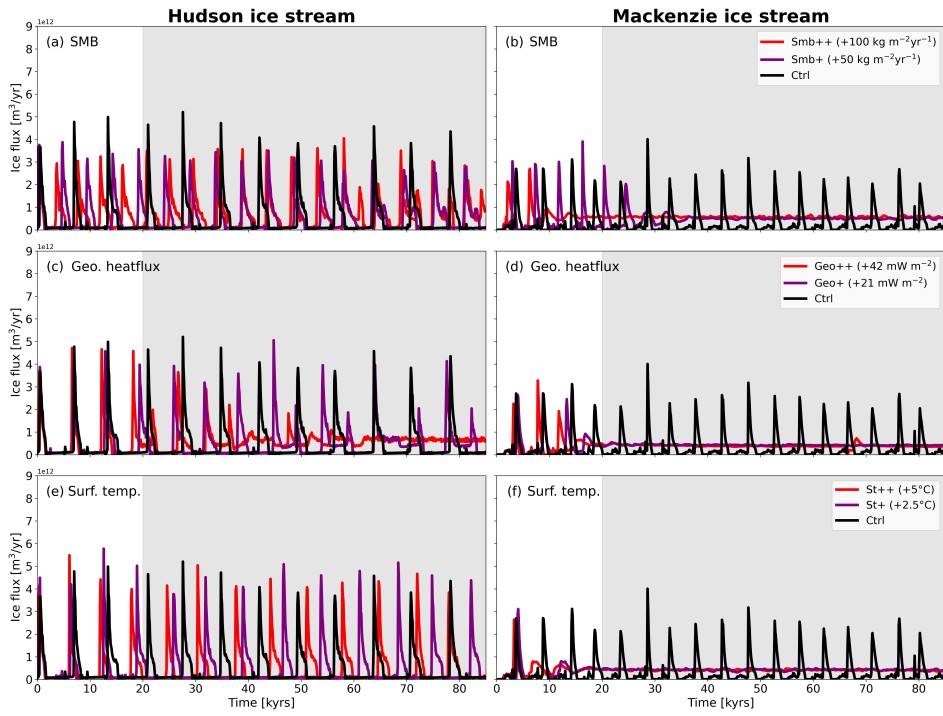

**Figure B1.** Similar to Fig. 7 but only shows the positive perturbation experiments where additional intermediate scenario simulations were performed. The panels show response to (a,b) SMB, (c,d) geothermal heatflux, and (e,f) ice surface temperature. Grey shading highlights analysis period.

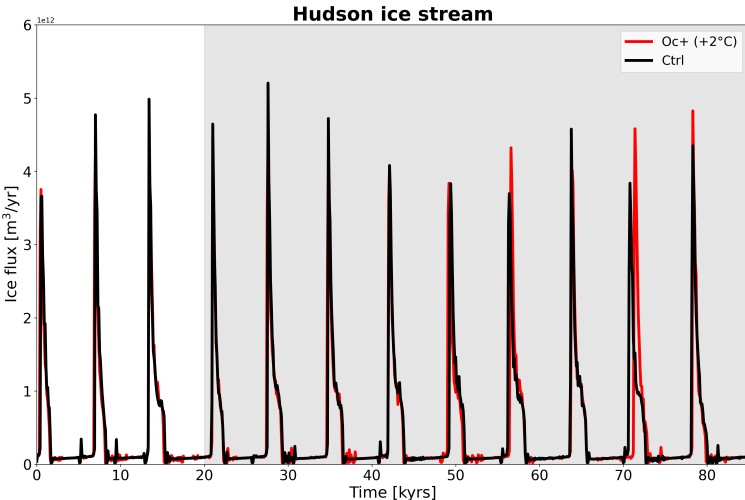

**Figure B2.** Timeseries of ice flux through the flux gate (Fig. 1) for ocean perturbation for Hudson ice stream.

*Code and data availability.* The PISM model code is publicly available through GitHub (https://github.com/pism/pism/, last access 01 April
2022). All scripts and required input data to reproduce all figures in the manuscript are available under the following link https://doi.org/10.
5281/zenodo.6519774.

*Author contributions.* CS and UM conceived the study. FAZ developed the setup with input from MLK. CS performed the experiments. CS
analysed the data. The manuscript was written by CS with input from all co-authors.

*Competing interests.* The authors declare that they have no conflict of interest.

*Acknowledgements.* C. Schannwell, M. Kapsch and U. Mikolajewicz were supported by the German Federal Ministry of Education and
Research (BMBF) as a Research for Sustainability initiative (FONA) through the PalMod project under the grant numbers 01LP1915C,
01LP1502A, and 01LP1917B. This work used resources of the Deutsches Klimarechenzentrum (DKRZ) granted by its Scientific Steering
Committee (WLA) under project ID ba0989. Development of PISM is supported by NSF grants PLR-1644277 and PLR-1914668 and NASA
grants NNX17AG65G and 20-CRYO2020-0052. We thank Andreas Wernecke for comments which improved the manuscript.

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
