# Peer review of "Sensitivity of Heinrich-type ice-sheet surge characteristics to boundary forcing perturbations"

_EGUsphere, 2022_

## Author Comment (AC1)

**Response to the reviewers**

We thank both referees for their thoughtful and thorough reviews of our paper. We appreciate you taking the time to complete these reviews and welcome your helpful comments. In the following we address their concerns point by point. Throughout this response to review document referee comments are provided in regular, non- italic font text, our response comments are provided in red font.

To accomodate the main comments of the reviewers, we have made the following main changes to the manuscript:

- We put more focus on the differences between the responses of the two ice streams to the different perturbations.

- We extended the simulation length by 27,500 years from 57,500 to 85,000 years, but only consider the last 65,000 years for analysis and the first 20,000 years as spin-up to alleviate the out-of-equilibrium effect.

- We expanded the methods section to provide more detail on the implementation of the surges in our model setup.
* * *
**Reviewer 1**

**Reviewer Point P 1.1** — This is an interesting paper that seeks to advance understanding of the North Atlantic's Heinrich Events (HEs), which are well documented sediment layers strongly suggesting large-scale periodic surges from the former Laurentide ice sheet. The paper is a significant advance in this field and will be important to communities working on paleoclimate as well as the ice-sheet instability. It investigates the impact of ice sheet external forcing and boundary conditions on surge characteristics by way of a series of sensitivity experiments using a standalone ice-sheet model. It contains a description and analysis is the basic surge mechanism from a control experiment, as well as analysis of paired (high-low) experiments in which the magnitude and frequency of external forcing is varied. The text is, on the whole, clearly written and structured, and the figures well presented and informative.

**Reply**: We thank the reviewer for the positive assessment of our manuscript. However, we would like to point out that for the majority of our simulations, we use a coupled ice sheet-solid earth model (see for example L58 in preprint).

**Reviewer Point P 1.2** — The authors should consider refocussing the Introduction to better reflect the remainder of their paper, which focusses on internally generated surges as opposed to mechanisms that rely on interactions with other components of the Earth system. This refocussing might be informed by the following suggestion.

**Reply**: We have expanded the paragraph on internally forced ice-sheet oscillations including appropriate references to give a more in-depth introduction to the main topic of the paper. We have decided to keep the paragraph on alternative hypotheses that have been suggested to trigger ice-sheet surges to give a

more complete overview of the field. Moreover, we have followed reviewer's 2 suggestion and removed the paragraph on the effects of Heinrich events on the climate system (L43-56). We have replaced this with a paragraph on internally driven mountain glacier surges that are relevant for the observed different surge behaviours of Mackenzie and Hudson ice stream.

**Reviewer Point P 1.3** — One of the more interesting aspects of the work is this discovery of two types of ice-sheet surging. One (Mackenzie) is closely allied to the original binge-purge concept of MacAyeal, while the other (Hudson) has a very different mechanism linked to the upstream propagation of sliding. Much more could be made of the differences between these two surge mechanisms, such as why they occur in their specific geographical locations (perhaps linked to channels in the subglacial topography) and the contrasting impact of changes in forcing on their dynamics.

**Reply**: We agree that this is indeed very interesting. We believe that the modelled behaviour is in large part affected by their different glaciological setting (marine-terminating vs. land-terminating). Further evidence for this is the fact that the surge behaviour of the land-terminating Mackenzie ice stream closely resembles that of present-day mountain glaciers and that of Hudson ice stream of present-day tidewater glaciers. In addition to adding more material to mountain glacier surges to the introduction, we have rewritten the section on the description of the different surge behaviours to highlight the differences in surge behaviour between the two ice streams.

**Reviewer Point P 1.4** — There are issues with the design of the anomaly experiments. I believe the anomalies are applied instantaneously at the start of each experiment so that there will be a long-term trend through the experiment as the modelled ice sheet gradually responds to the often large changes in forcing. These experiments may not therefore measure the full impact of the anomaly; this would require a separate spin-up for each anomaly experiment. This needs to be made very much clearer in the text and the implications for the results discussed. Alternatively, the anomaly experiments could be rerun with appropriate spin up.

**Reply**: The reviewer is correct in the assumption that the anomalies are applied instantaneously. We agree that a spin-up is necessary to truly measure the impact of applied forcings on the surge characteristics. Based on the ice volume trend in the anomaly experiments, we decided that a 20,000 year spin-up is appropriate. We therefore extended all simulations by 27,500 years to 85,000 years, but only consider the last 65,000 years for analysis.

**Reviewer Point P 1.5** — The surging behaviour observed is not regular and exhibits variations in duration of active/quiescent period, as well as magnitude, in a single experiment. This means the results need to be treated very carefully. The situation is compounded by the small sample size of surges within each experiment (typically 10), which makes statistical analysis difficult. It would be interesting to investigate whether a series of experiments with a particular type of anomaly (for instance varying from Smb- through Ctrl to Smb+) would support the conclusions drawn here or whether surge irregularity and sampling issues would create a more complicated picture. Finally, the ranges chosen for each pair of anomaly experiments seem fairly arbitrary so that it is problematic to compare their impacts on surging (apples and oranges issue). Linking these ranges to the variability exhibited in the original MIS3 experiment may be one way of tackling this issue.

**Reply**:   To sample the parameter space more densely, we performed three more simulations in the positive anomaly range for the SMB, geothermal heatflux, and surface temperature variables. We also compare the tested ranges for surface temperature and SMB with the model spread shown in the PMIP4 LGM ensemble. This comparison shows that our tested ranges for surface temperature and SMB are much smaller than the spread shown in the PMIP4 LGM ensemble.

**Reviewer Point P 1.6** — The discussion concerning phase locking and synchronicity (both with other elements of the Earth system and between individual ice streams) is a little naïve. To my mind, there are genuine questions concerning whether phase locking is possible given the small number of surges/oscillations possible within the lifetime of the Laurentide during a single glacial period. None of the experiments presented unequivocally demonstrates phase locking (which would require for instance the phasing of the two ice streams' surges to evolve toward synchroneity through time) and much of the discussion is therefore speculative.

**Reply**:   Given that some observational records indicate synchronous surges from different ice sheets (Grousset et al. 2000), we believe this is a hypothesis worth exploring. Previous model results, admittedly using simpler model setups, have demonstrated phase locking within ∼10 kyrs (Calov et al., 2002, Kaspi et al., 2004, Mann et al. 2021). However, our main point is that in our simulations no phase locking occurs. The only possible avenue to simulate synchronous ice-sheet surges from Mackenzie and Hudson ice stream is to manipulate the forcing such that surge periods are similar, but not through a common external trigger. We have amended the text to make this more clear.

**Reviewer Point P 1.7** — The discussion on the interaction of the two surging ice streams by migration of the ice divide is interesting but lacks detail on the mechanisms involved. In particular, it is unclear how this type of interaction could occur given that the time taken for ice to flow from the divide to an ice stream is well in excess of the period of the modelled surges.

**Reply**:   The ice streams are closer together than that (ca. 500 km) and ice velocities in this branch are more on the order of 25-40 m/yr, so that this number is <10 kyrs. We hope that the video of Ctrl that we have added, makes this more clear. Further evidence is provided by Fig. 11, where the surge behaviour changes ∼10 kyrs after the start of the divide migration and the resulting growth of the Hudson drainage basin.

**Minor**

**Reviewer Point P 1.8** — L18: The Introduction suggests that there are two main schools of thought on the cause of HEs: internal-generated ice sheet oscillations and oscillations that rely on the interaction of the ice sheet with other components of the Earth system. The paper focuses of the former and is based around a model that exhibits internal oscillations of this type. The Introduction, however, mainly focusses on interactions. This seems odd and the Introduction would benefit from refocussing toward a fuller discussion of the literature around internal oscillations.

**Reply**: We agree and have rewritten parts of the introduction (see answer to P 1.2).

**Reviewer Point P 1.9** — L19: "implicated" need to make the link between the sliding perturbation of preceding sentence and ocean warming. Why do changes in sliding necessarily suggest that ocean forcing is important?

**Reply**: We added that this sliding perturbation mimics the effect of a rise in sea level and indicates that the ocean could possibly trigger ice-sheet surge events.

**Reviewer Point P 1.10** — L20: "only" see comment about strengthening review of literature on internal oscillation mechanism. For instance, Roberts et al. 2016 (Climate of the Past) perform a similar sensitivity analysis. There are also several papers by Marshal and Clarke that are also relevant.

**Reply**: We have added the citations and reworded the corresponding sentence.

**Reviewer Point P 1.11** — L21: My understanding of the Holland Jenkins (1999) three-equation model is that they use the temperature of the ocean sublayer immediately under the ice shelf as opposed to the ambient ocean temperature. How is this sublayer temperature found?

**Reply**: We have added an explanation of how this temperature and salinity is found. It reads: "The ocean temperature and salinity fields are provided by MPI-ESM. Because MPI-ESM does not resolve the cavities under ice shelves, we extend salinity and temperature fields to ice shelves through extrapolation. This 2D extrapolation is based on temperature and salinity averages between ∼200-400 m depth of the ocean model."

**Reviewer Point P 1.12** — L22: The spatial distribution of basal heating over a wider area is crucial to the propagation of the surges so that this detail is likely to be very important to the results reported elsewhere in the paper. The authors should explain and justify this choice and, if possible, describe its impact on their results.

**Reply**: We have added that this leads to a faster propagation of the surge front.

**Reviewer Point P 1.13** — L23: "better" this Is very subjective. In what way is the simulation better?

**Reply**: We reformulated this.

**Reviewer Point P 1.14** — L24: Need to say how long the experiments were? I guess 60 kyr based on the figures but would be good to say so.

**Reply**: We have added this information. Now simulations are 85,000 years long of which the first 20,000 years are considered model spin-up and the remaining 65,000 years are used for analysis.

**Reviewer Point P 1.15** — L25: Hudson ice stream is marine terminating. The movement of its grounding lines may therefore be important. Please include information on how the grounding line was modelled.

**Reply**: We added that the flotation criterion is used without additional flux conditions.

**Reviewer Point P 1.16** — L26: The initial state was taken from a transient simulation (MIS3) so presumably is not in equilibrium with the climate forcing at that time (36 ka in MIS3). Is this an issue and is there any evidence of model drift in additional to the internal oscillations?

**Reply**: We added a sentence that this period was purposely selected because global ice volume remained stable during the MIS3 period apart from the cyclic ice-sheet surges. This is corroborated by the fact that the ice volume change in the Ctrl simulation is <3% for the Laurentide ice sheet across the entire simulation.

**Reviewer Point P 1.17** — L27: It would be good to summarise the forcing strategy here as well. My understanding is that forcing for the period 36 ka to 23 ka (36 ka minus 13 kyr) is taken from the MIS3 simulation as a starting point and then adapted by taking the temporal mean or applying anomalies etc. This 13 kyr (as two 6.5 kyr cycles) is then repeated for the duration of the experiment (60 kyr).

**Reply**: We have added a paragraph with the motivation of why we construct the composite climate forcing. The main objectives are to reduce the effect of climate variability not associated with ice-sheet surges and secondly, to construct a forcing with minimal temporal trend that can be applied as many times as desired, and hence requires a smooth transition from one forcing cycle to the next.

**Reviewer Point P 1.18** — L28: "favourable" this is again subjective. Please define what is meant by this and explain why it is important.

**Reply**: This has been dropped in the rewritten section.

**Reviewer Point P 1.19** — L121-129: I found this description confusing. Figure 2 helps but I think there is room for a clearer explanation. Perhaps it would be helpful to explain why this seemingly complicated process is necessary.

**Reply**: We have added a paragraph why we think this forcing procedure is necessary and also rewrote the description of the forcing generation.

**Reviewer Point P 1.20** — L139: 3D fields. Not clear to me what is meant here. Surface temperature and SMB are inherently 2D fields. Is the 3rd dimension time or vertical dependence (perhaps used in an interpolation downscaling scheme)?

**Reply**: Yes, the EBM calculates these fields at prescribed elevation levels which has computational advantages and allows to interpolate the fields onto any ice-sheet topography. For more information on this see Kapsch et al. 2021, TC. We removed 3D from the text and just state that these fields are downscaled from the EBM output.

**Reviewer Point P 1.21** — L140: Ocean forcing. This relates back to the use of the three-equation model.

**Reply**: see response to P 1.11

**Reviewer Point P 1.22** — L141: Is the same initial condition used in the anomaly experiments as the control? This implies that the forcing (after anomaly applied) will be out of equilibrium with the initial state (even more so than the issue on line 119). Is this a problem? Presumably there will be an overall trend in the anomaly experiments as the ice sheet responds. How does this

relate to the internal oscillations? I don't see much evidence that their character changes through each individual experiment.

**Reply**: Yes, the same initial conditions are used. To alleviate the out-of-equilibrium effect, we have extended the simulations to 85,000 years, and only use the last 65,000 years for analysis and regard the first 20,000 years as spinup (see reply to P 1.4).

**Reviewer Point P 1.23** — This also raises the issue of what actually is being done in these anomaly experiments. If the control initial condition is being used in the anomaly experiments, the anomalies may not have enough time during the experiment to affect much change (to say ice thickness or internal temperature field). The analysed impact of a particular anomaly might therefore be its instantaneous impact as opposed to its larger, longer-term impact.

**Reply**: We agree. See reply to P 1.4.

**Reviewer Point P 1.24** — Table 1. Unclear what "experiment type" refers to.

**Reply**: This has been added to the table caption.

**Reviewer Point P 1.25** — L147: I think it would be worth being even more explicit about the forcing here: there is no temporal variation in forcing so that any surging behaviour observed must be due entirely to internal mechanisms.

**Reply**: We added that the forcing is "time constant" to make this more clear.

**Reviewer Point P 1.26** — Figure 3. Why is the unit m2/yr? Is this the average flux through the gate per m width? I would have expected m3/yr.

**Reply**: The reviewer is correct. There was a mistake in the plot script. Corrected here and for all other Figures.

**Reviewer Point P 1.27** — L163: This is one of the most interesting parts of the paper – the difference types of surge in the Mackenzie and Hudson ice streams. The authors should consider comparing their surge mechanism with the paper that first described this style of surging (Payne et al 1995 JGR) which contains a more process-based analysis of the mechanism than Calov et al. (2002) including the roles of ice temperature and stress concentrations in the initiation and termination of a surge.

**Reply**: We thank the reviewer for pointing us to this relevant paper. We rewrote the section on the description of the different surge behaviours. Indeed the surge behaviour of Hudson ice stream closely follows the surge style described in Payne et al 1995 JGR. This link has been added to the surge description.

**Reviewer Point P 1.28** — This style of surge is related to but distinct from the original binge-purge theory of MacAyeal (1993). In the latter, the bed of the ice sheet interior warms leading to a pocket of fast flow that propagates towards the margin. In the former, stress concentrations at interface between fast and slow flow leads to local warning and the upstream migration of this

interface. It is fascinating that both types of surge are found in different parts of the ice sheet, and the text could usefully be refocussed around discussing this distinction.

**Reply**: Yes, we make it now more explicit that Hudson ice stream follows the mechanism of stress concentrations at the interface between fast and slow flow. To support this we also added driving stress to the flowline plots, which nicely illustrates this point. We also link the observed behaviour for Mackenzie and Hudson ice stream to present-day mountain glacier surges which show a very similar surge behaviour.

**Reviewer Point P 1.29** — L169: Grows is imprecise – is it ice thickening or later extension that is important?

**Reply**: Agreed and removed.

**Reviewer Point P 1.30** — L170: It is not clear why delta stress (driving stress minus basal shear stress) is a meaningful metric. Heat generation by friction within the ice sheet and at its bed are both functions of stresses and strain rates. I don't see how delta stress helps in this analysis.

**Reply**: We agree and have removed delta stress from the plot and condensed the layout to a 3x2 Figure, but added driving stress. The same applies to Figure 6.

**Reviewer Point P 1.31** — The links between this description (163-177) and Figure 4 could be tighter (only three of the panels are referenced).

**Reply**: This has been tidied up.

**Reviewer Point P 1.32** — Figure 4. X axis title and labelling missing.

**Reply**: Apologies. This has been fixed.

**Reviewer Point P 1.33** — L178: Not clear what "ice-stream front" refers to. Marine margin? Use of the term front also crops up later as well and should be tightened (they are a number of 'fronts' of different types in the model).

**Reply**: Changed to "marine margin".

**Reviewer Point P 1.34** — L179: It don't think it is clear from Figure 6 that propagation is downstream, in fact panel e suggests warming at the margin before warming in the interior. This might be a case of selecting a better time for the pre-surge time slice. Also giving times for each phase would help, i.e. what is the time difference between panels e and h?

**Reply**: We have added the time information to Figures 4 and 6. The development of a full-blown surge is characterised by a downstream propagation of the surge. However, Mackenzie ice stream in between big surge events is also characterised by small-scale accelerations and periods of advance that never evolve into a full surge, because the ice upstream is not preconditioned to surge yet (still frozen to the bed).

**Reviewer Point P 1.35** — Here again it would be worth emphasizing that this mechanism is closer to the original binge-purge concept. Also worth mentioning that Mackenzie is land terminating while Hudson is marine terminating.

**Reply**: This has been added to the rewritten section.

**Reviewer Point P 1.36** — L184: I don't think there is enough evidence to state that the warmer air temperatures are the "likely" cause. Can you exclude other components of the heat budget such as variations in geothermal heat flux and/or reduced vertical advection?

**Reply**: We changed this to "could be caused"

**Reviewer Point P 1.37** — Figure 5. Why does surface gradient have units of m-1. Should this not be nondimensional? Need to clarify relation between distance on x axis and Figure 1 (i.e., is it measured from upstream or downstream end). Similarly, clarify what "ice stream front" means. Also need to explain and justify the choice of surface gradient as a metric. Is this because of its role in controlling gravitational driving stress/frictional heat production? If it is simply as a map of the extent of the surge then why not use ratio of surface to basal velocity (of just the surface velocity itself).

**Reply**: The reviewer is correct that the surface gradient should be dimensionless. We corrected this in the revised MS. We feel that the upstream and downstream labels as well as the reference to the flowline location in Fig. 1 are sufficient info for the reader as to where the Hovmoeller diagram is located. We replaced "ice-stream front" with "marine margin". We indeed chose the surface gradient because of its role for gravitational driving stress as well as it illustrates the propogation of the surge front nicely.

**Reviewer Point P 1.38** — L196: The ranking of different boundary forcing anomalies seems arbitrary. The individual +/- experiments are interesting but it is hard to compare them to make statements about which type of forcing is more important. Some attempt to justify the anomaly range is made for each forcing variable, however I do not think this is precise enough to allow comparison between them. For some of the perturbations it would make sense to make the +/- range a fraction of the variability seen in the MIS3 simulation (e.g., SMB, surface temperature, sea level, ocean temperature), although this would not work for geothermal heat flux.

**Reply**: We agree that it is indeed challenging to rank the importance individual variables on the surge cycle. We therefore focus our analysis now more on the differences between the responses to the different pertubations for the two investigated ice streams. Using fractions instead of absolute values is a valid alternative, but it would be computationally too expensive to rerun the entire ensemble. To put our tested ranges into better context, we added a comparison of the model spread for SMB and surface temperature shown in the PMIP4 LGM ensemble.

**Reviewer Point P 1.39** — L196: Another issue with this comparison is that the surges are at least in part chaotic, i.e. the magnitude and active and passive duration of individual surges is not the same even in one experiment. The length of each experiment is also short compared to a surge so that each contains only 10 surges. It is therefore difficult to compare details. Performing a number of experiments in each +/- forcing range would shed light on this issue.

**Reply**: We have extended the simulations by 27,500 years and added three more simulations in the $+$ range to the anomaly ensemble to make the comparison more robust.

**Reviewer Point P 1.40** — Figure 4 and 6. Need an indication of the time for each phase so that they can be related to Figure 3.

**Reply**: This has been added.

**Reviewer Point P 1.41** — L214: Is the geothermal heat flux a constant? Need to clarify.

**Reply**: This is mentioned in the methods section where it reads: "For the forcing of the temperature equation, we apply a time and space invariant geothermal heat flux of 42 mW m$^{-2}$, if not stated otherwise."

**Reviewer Point P 1.42** — L215: You need to be very careful with statements such as "most sensitive to changes in SMB" this really assumes that all of the perturbations have the same 'strength', which I do not think you are able to do. The only way of doing this would be to use a common metric, i.e., a fraction base on their variability (see 196).

**Reply**: We have removed this statement and focus now more on the difference in the response to the different perturbations between the two investigated ice streams.

**Reviewer Point P 1.43** — Figure 8. I estimate that each experiment contains around 10 surges so that illustrating their variability using standard deviation is overkill. Why not show all individual surges in an experiment as a cloud of separate points (with same colour)? Also need to indicate how period is calculated/defined. Difference in time between maximum and minimum ice volume?

**Reply**: We added how the surge period is defined. We tried the suggestion of the reviewer, but came to the conclusion that this would make the Figure to cluttered (see below).

**Reviewer Point P 1.44** — L265. Worth stressing here that the ice streams have two very different surge mechanisms hence forcing anomalies likely to affect them in different ways.

**Reply**: This has been added.

**Reviewer Point P 1.45** — L266. Comparison between the present day and LGM is not a good a measure of the validity of these anomalies (the Laurentide ice sheet does not exist at the present day). Better to compare against observed or modelled changes for times when the ice sheet existed at roughly the size that you are simulating (i.e. variability during the glacial period).

**Reply**: We agree with this and have added a plot to the supplementary material that shows the maximum ensemble spread of PMIP4 models for surface air temperature and precipitation (as proxy for surface mass balance). The plot highlights that variability across the participating models is much larger than the ranges we tested for surface mass balance and surface air temperature (see Fig. below).

**Reviewer Point P 1.46** — L267. This may also link to the saddle collapse mechanism of Gregoire et al. (2012).

[Figure]

**Reply**: We have added a citation to this paper.

**Reviewer Point P 1.47** — Figure 9. These plots strongly suggest that there is a trend in the model's response in the anomaly experiments (i.e., that the anomaly is imposed instantaneous at the start of the experiment). As indicated elsewhere, the ice sheet will require time (thousands of years) to achieve its full response to all of the anomalies by either evolving its internal temperature field or its geometry. The use of instantaneous anomalies does not therefore measure the full response of the ice sheet to the anomaly. For instance, increased surface air temperature by itself will have little immediate effect but might have a strong effect after this warming has had time to propagate down through the ice to impact basal thermal regime. The same is true of all of the anomalies with the possible exception of geothermal heat flux (because it is applied at the bed), although even here there is likely to be a difference between the instantaneous and long-term response.

**Reply**: Yes, we agree. That is why we extended the simulations by 27,500 years (see also reply to P 1.4).

**Reviewer Point P 1.48** — Figure 9. Would be good to include experiments in figure caption (also for Figures 11 and 12).

**Reply**: We think that having the experiment names in the figure legend is suffcient.

**Reviewer Point P 1.49** — L292: Give logic for why this was only done for SMB and GIA.

**Reply**: We added the motivation of why we chose these two single forcings to the text. In brief, we selected GIA because we did not test its influence in the anomaly experiments and we selected the SMB because it showed a strong influence on the surge cycle length in the anomaly experiments.

**Reviewer Point P 1.50** — L293: My understanding of phase locking is that many oscillations are required for it to develop. I do not see how it can happen given the small number of ice sheet surges that are possible during a glacial period. The oscillations reported here have period 10 kyr so at most 10-20 oscillations are possible during the lifetime of the Weichselian Laurentide ice sheet. Phase locking would require a mechanism that can continue through interglacials in the absence of the ice sheet.

**Reply**: See reply to P 1.6.

**Reviewer Point P 1.51** — L294: Comments about the number of oscillations required for phase locking are also relevant here (even more so since synchronisation of surges can definitely only happen while the ice sheet exists).

**Reply**: Yes, we have added this information to the text.

**Reviewer Point P 1.52** — L295: Potential interaction between the Hudson and Mackenzie by ice divide migration is interesting, however the present discussion is a bit simplistic and ignores the time required for a change in upstream catchment area to impact the ice flow within the ice stream. The ice streams are typically 1000 km apart (Figure 1) suggesting a travel distance of 500 km from divide to ice stream. Given ice velocities of ∼10 m/yr upstream of the ice streams

(Figures 3 and 6), a travel time of ~50 kyr is required. This is similar to the period of the surge limiting the potential for interaction.

**Reply**: The ice streams are closer together than that (ca. 500 km) and ice velocities in this branch are more on the order of 25-40 m/yr, so that this number is <10 kyrs. We hope that the video of Ctrl that we have added makes this more clear. Further evidence is provided by Fig. 11, where the surge behaviour changes ~10 kyrs after the start of the divide migration and the resulting growth of the Hudson drainage basin.

**Reviewer Point P 1.53** — L296: This paragraph is speculative. Finding two experiments (of 17) that have the same phase is not evidence for the possibility of phase locking. Stronger evidence could come from a series of experiments based on Geo- and St+ that show the development of locking (i.e., oscillations that are initially out of phase but become locked). This also relates to conclusion on line 382.

**Reply**: Yes, the phrasing was not precise in this paragraph. Our point is that we cannot get a phase locking in our simulations and can only simulate surges that appear synchronised when the correct combinations of boundary forcings are used. We have rephrased this here and elsewhere to make this more clear.

**Reviewer Point P 1.54** — L297: More generally, plotting the ice speed (or flux) for Mackenzie and Hudson flux gates against one another would be a good way of displaying potential synchronicity. This type of phase diagram would show synchronicity as a closed loop.

**Reply**: See reply to previous point.

**Reviewer Point P 1.55** — L298: "positive climate perturbation" is hard to understand – needs to be more specific.

**Reply**: Changed to "positive perturbation in the anomaly experiments"

**Reviewer Point P 1.56** — Technical corrections

**Reply**: All fixed.
* * *
**Reviewer 2**

**Reviewer Point P 2.1** — This authors present experiments performed with an ice-sheet model coupled with a GIA model over the North American domain pertaining to representative MIS-3 boundary conditions. As built, the ice sheet exhibits very large magnitude, quasi-periodic oscillations primarily in the Hudson Strait ice stream and the Mackenzie ice stream that are supposed to correspond to Heinrich Events. Along with an in-depth analysis of the dynamics of these ice surges, different purturbations in the boundary conditions are tested to assess their influence on the characteristics of the surges. The topic is interesting and open in the literature, and the contribution here with a state of the art model setup is valuable. The paper is also well written and

flows nicely. Nonetheless, I believe framing of the context of these experiments needs to be refined and the credibility of the model setup producing the surges themselves needs more support.

**Reply**: We thank the reviewer for the positive assessment of our manuscript.

**Reviewer Point P 2.2** — Framing. The Abstract and Introduction is very nicely written. However, the description of Heinrich events here based on strong assumptions resulting from the authors' own modeling work. Not all characteristics are known to be true, and thus should not be presented as definitively representative of reality. This is particular of the sentence on L3 in the Abstract and the paragraph on L43-56 in the Introduction. It is well known that the AMOC collapse related to Dansgaard-Oeschger Events occurs prior to Heinrich Events in sediment records (e.g., Barker et al., 2015, http://dx.doi.org/10.1038/nature14330). In other words, glacial stadials and the dominant climatic impacts recorded globally, are mainly related to ocean circulation changes. It may be that freshwater discharge during Heinrich Events can act to extend the duration of the stadial, but this is far from clear in the paleo records. Thus, discussing all of the "impacts" of Heinrich Events is not really relevant, at best, and rather misleading at worst. I would simply recommend removing this sentence and this paragraph from the manuscript, as it is not necessary for motivating the study of Heinrich Events, which are large scale and enigmatic in any case.

**Reply**: As suggested we have removed the corresponding text.

**Reviewer Point P 2.3** — Another example is the phrase "during which large amounts of ice are discharged" on L2 in the Abstract. We have no constraints from reconstructions on the total ice mass that was discharged during HEs. Some modeling studies do not show such large fluctuations in volume (e.g. Alvarez-Solas et al., 2013), while the model the authors use does. So, again, I would suggest more precision when discussing what we know versus what comes out of this work. This will facilitate putting the results in context of the open questions on this topic, and will increase the value of the work.

**Reply**: While we agree there are large uncertainties regarding the amount of ice that was discharged during Heinrich events, we think that the current scientific consensus is that Heinrich events are associated with large ice-sheet discharge events (e.g. MacAyeal 1993 , Marshall and Clarke 1997, Hemming 2004, Roberts et al. 2014, Bassis et al. 2017, Ziemen et al. 2019). However, to be more precise we changed this to "large numbers of icebergs are released from the Laurentide ice sheet"

**Reviewer Point P 2.4** — Experimental setup. The model components used appear to be state of the art and include many/ most processes one would be interested in for studying this problem. However, the results exhibit ice velocity values that sound simply incredible. In the surge phase, it appears that maximum values of up to 40,000 m/yr are reached, with inland values over a distance of 2000 km of $> 1,000$ m/yr (see Fig. 4, bottom-center panel and Fig. 5b). This seems to me extremely implausible, and at the very least should be acknowledged as such in the text, or explicitly justified and explained.

**Reply**: Given that observations for a mountain glacier surge (Variegated Glacier, Alaska) have shown glacier speeds of 22,000 m/yr (or 60 m/d), we do find our velocity values reasonable. Many Antarctic ice stream have ice velocities faster than 1,000 m/yr. Our modelled ice velocities are higher than what

has been reported in other studies (e.g. Alvarez-Solas et al., 2013), but as pointed out by the reviewer large uncertainties about the characteristics of these events remain.

**Reviewer Point P 2.5** — Furthermore, I think such a result is predominantly dicated by the choices regarding basal friction, which are only described in a very cursory way here. In particular, I think the reader should see more information about how $\tau_c$ is calculated in the basal friction law, as well as the relation used to calculate effective pressure, which I assume is an input to $\tau_c$. Rather than "basal sliding is discouraged", I think actual values should be given and any spatial maps used should be included here. The point is, not only do I think this is valuable information for understanding the nature of the surges presented here, but all information should be provided to allow others to reproduce the results.

**Reply**: We agree that the basal sliding parameterisation has a strong influence on the surge behaviour and that our description of how this is implemented in our simulations was lacking detail. We therefore expanded this section of the model description and provide more detail on how basal sliding is parameterised in the presented simulations. We also added the evolution of basal friction to our supplementary movie of the reference simulation.

**Reviewer Point P 2.6** — I am also surprised by the choice of a constant value of GHF of 42 mW/m2 everywhere. In the motivation for the range of the Geo+/- experiments, the authors cite a spatial GHF map (Lucazeau, 2019) of which several exist. Why not apply such a map as the default distribution and scale from that?

**Reply**: The value of 42 mW/m2 has been used frequently in model studies of northern hemispheric ice sheets (see for example Näslund 2005, Annals of Glaciology). While we agree that a spatially varying geothermal heatflux would be more appropriate, we would also like to highlight that there are large uncertainties associated with these maps. This is also visible if different geothermal heatflux maps for Antarctica from the last 10-15 years are compared. Moreover, the goal of our simulations is not to produce the most realistic setup possible, but investigate the response of our reference simulation to various perturbations. Of course parameter values should be located in a realistic range. However, we believe a GHF of 42 mW/m2 is within such a range.

**Reviewer Point P 2.7** — Specifically with regards to Geo-, GHF values of $<\sim40$ mW/m2, and particularly of 0 mW/m2 do not seem likely to exist in reality. The motivation of "Geo-" is therefore difficult to understand. Since a very low value of GHF is imposed by default, I would expect only a "Geo+" purturbation to be needed.

**Reply**: We agree that this value is rather unlikely to exist. We chose this value to have symmetric anomaly experiments for all considered variables. In the revised MS, we have also added an intermediate simulation that used 63 mW/m2 as GHF to investigate some intermediate scenarios to the runs presented in the preprint, as was also suggested by reviewer 1.

**Reviewer Point P 2.8** — With regards to setting the imposed value of GHF to -1000 mW/m2, I am very surprised that this did not cause numerical problems for the model. It seems that the authors were able to achieve their goal of deactivating streaming, but out of curiosity, what does the vertical temperature profile of ice look like in these regions? To keep conditions within more

realistic bounds, the authors may consider simply modifying the spatially-defined friction coefficient in these regions for these tests.

**Reply**: Please find below a plot of the temperature profile at ∼570 km of the Hudson stream flowline for various time slices. We agree there are alternatives to implement this and a smaller negative geothermal heatflux might have also worked. However, for values down to ∼-100 mW/m2 it took >15 kyrs to shut off the surges. Because of this and to reduce computation time, we decided to apply a larger heatflux.

[Figure]

**Reviewer Point P 2.9** — Finally, I think overall some additional figures would help. The readers should see some maps of the whole ice sheet being simulated in Ctrl (surface elevation, velocity, basal temperatures, $\tau_b$) and possibly in the different phases (Quiescent, Pre-surge, Surge). For example, a supplementary movie, showing the surge behavior in the control simulation would be very valuable as well.

**Reply**: We agree and have added a movie of the Ctrl simulation in which we show the evolution of ice speed, ice surface elevation, driving stress, and basal shear stress.

**Specific comments**

**Reviewer Point P 2.10** — L1 and L18: "among the most dominant" <= What makes them most dominant? Consider rephrasing simply to "prominent"

**Reply**: Fixed.

**Reviewer Point P 2.11** — L35: More recent theories propose => A more recent theory proposes

**Reply**: Fixed.

**Reviewer Point P 2.12** — L38: tidewater glacier => tidewater glaciers

**Reply**: Fixed.

**Reviewer Point P 2.13** — Paragraph L43-56: As mentioned above, this paragraph gives a misleading characterization of HEs. Climate impacts are largely due to AMOC shutdown. In addition, MWP1a is a very specific event during the deglaciation (L53-56). It cannot be considered as representative of Heinrich Events in general. Thus citing it here is not really appropriate

**Reply**: This paragraph has been removed.

**Reviewer Point P 2.14** — L74: cylce => cycle

**Reply**: Fixed.

**Reviewer Point P 2.15** — L106: 36 ka => 36 ka ago [or before present?]

**Reply**: Changed to before present.

**Reviewer Point P 2.16** — L109: bathymetrie => bathymetry

**Reply**: Fixed.

**Reviewer Point P 2.17** — L164: Consider also citing Feldmann and Levermann (2017), as they describe this mechanism in detail with experiments using PISM.

**Reply**: We have rewritten this section and have added this citation.

**Reviewer Point P 2.18** — L184-185: What about the impact greater thickness, insulating and warming the base upstream? This seems like it would also contribute significantly.

**Reply**: Yes, we have added this.

**Reviewer Point P 2.19** — L214-215: Map of Lucazeau (2019) only goes down to values of about 40 mW/m2.

**Reply**: Our interpolated GHF map from Lucazeau goes down to values of 25 mW/m2 for the Hudson region, and to values of 30 mW/m2 for the Mackenzie region. To acknowldege this, we rephrased this sentence to "This choice is in a similar range to that of geothermal heatflux maps of present-day North America (Lucazeau, 2019)."

**Reviewer Point P 2.20** — L252-253: It is hard to compare magnitude sensitivity, given that the units are totally different. Consider removing this sentence, as it is not necessary, or at least

reformulating. Perhaps you can quantify the sensitivity in terms of $\Delta t$ surge time as a function of percent change in boundary variable. Maybe this would allow a more appropriate comparison.

**Reply**: This sentence was removed during the rewriting of this section as we shifted the focus more towards the difference in the response to the different perturbations between the investigated ice streams.

**Reviewer Point P 2.21** — L258: sensitive => sensitively

**Reply**: Fixed.

**Reviewer Point P 2.22** — L293: What kind of frequencies are imposed on GHF simulations? Are these not constant? Or rather this is GHF at the based of the ice sheet, which can evolve transiently, as opposed to GHF deep in the bedrock? Please clarify here more explicitly.

**Reply**: We added the information that the GHF is specified at the bottom of a 1 km thick bedrock layer to the methods section.

**Reviewer Point P 2.23** — L298: sensitive => sensitively

**Reply**: Fixed.

**Reviewer Point P 2.24** — L367-370: Again, this statement may be true, but it cannot be justified by the units used. Is a perturbation of +100 mm/yr of SMB equivalent to a surface temperature increase of +5 K? Find a way to rephrase, or rescale.

**Reply**: We rephrased this in the rewrite of the conclusion section.

---

## Author Response (AR2)

**Reviewer 1**

**Reviewer Point P 1.1** — I find the revised version of the manuscript much improved, and the additional figures and video are valuable. I still have serious doubts about the realism of such large velocities propagating on millennial timescales over such large distances into the ice-sheet interior. However, I would not be opposed to publication. The origin of Laurentide ice sheet surges is still an open question, and these results can provide further insight to the community. I only have one more general comment below that I would like to see addressed, as well as one very minor specific comment.

**Reply**: We are glad for the positive assessment from the reviewer of our revised manuscript.

**Reviewer Point P 1.2** — I appreciate the reference to surging glaciers, and that such magnitudes of velocities (10s of kilometers per year) can be reached in reality. However, these glaciers span no more then several kilometers, not hundreds of kilometers, and the surges are usually seasonal. Thus, the analogy may break down. My doubts are further enhanced when looking at Figure 4 or Figure 6. The surface elevation of the ice sheet appears to become quite noisy, particularly at higher velocities. This is apparent in the curves of driving stress in the right-hand panels. I find this noise very unusual, as the ice sheet tends to act as a smoother to any perturbative stresses. In my experience, this is more indicative of numerical instability of the model. So I would ask that the authors confirm that this reflects a physical phenomenon as opposed to numerical noise. If it cannot be fully ruled out to be numerical, perhaps this is fine, but in either case, this surprising behavior should also be commented on somewhere in the text.

**Reply**: We agree that the modelling of the surges due to the high velocities are numerical challenging and impose high stability demands on the employed linear solvers. However, there is no evidence of numerical instability in either the stress balance solution or the ice thickness evolution solution in any of our simulations as both solvers always converge. The time stepping criterion in our PISM version is additionally even more stringent than in the base version to exactly avoid the situation raised by the reviewer. We have added a sentence to the revised manuscript that acknowldeges the numerical challenging nature of the Heinrich events. It reads: "The modelled high ice velocities and large velocity gradients during the surges pose a formidable challenge for the model numerics. However, in all presented simulations, solutions to the stress balance and ice thickness evolution equation always converge, confirming the robustness of our results."

**Specific comments**

**Reviewer Point P 1.3** — L115-120: It would be more intuitive to frame this in the opposite way, by stating that "... basal sliding is further enhanced in regions where sediment is present ..."

**Reply**: Agreed. Changed accordingly.